# Programmable spatial magnetization stereolithographic printing of biomimetic soft machines with thin-walled structures

Xianghe Meng [1,2], Shishi Li [1,2], Xingjian Shen[1], Chenyao Tian [1], Liyang Mao [1] & Hui Xie [1] ✉

Soft machines respond to external magnetic stimuli with targeted shape changes and motions due to anisotropic magnetization, showing great potential in biomimetic applications. However, mimicking biological functionalities, particularly the complex hollow structures of organs and their dynamic behaviors, remains challenging. Here, we develop a printing method based on three-dimensional uniform magnetic field-assisted stereolithography to fabricate thin-walled soft machines with internal cavities and programmable magnetization. This printing technique employs Halbach arrays and an electromagnetic solenoid to generate an adjustable uniform magnetic field (up to 80 millitesla), efficiently orienting ferromagnetic particles, followed by solidification with patterned ultraviolet light. A support strategy and optimized material composition enhance printing stability and success rates. Our developed method enables fabrication of magnetic-driven soft machines capable of peristaltic propulsion, unidirectional fluid transport, periodic pumping action, and intake-expulsion deformation. These structures, achieving hollow ratios as high as 0.92 and enabling parallel manufacturing, highlight this technique's considerable potential for biomedical applications by emulating complex biological behaviors and functions.

Soft machines hold substantial potential for applications in biomimetics[1–4], healthcare[5–8], and robotics[9–12] due to their inherent flexibility, adaptability[13,14], and biocompatibility[15,16]. In biomimetics, the aim is to translate the structures, behaviors, and functions of biological organisms into machine design and actuation principles, enabling performance similar to natural systems[17–19]. Common actuation mechanisms for biomimetic soft machines include fluidic[8,10], electrical[9,20], magnetic[11,12], chemical[21,22], optical[23], and thermal[24] methods. Among these, magnetic actuation is particularly advantageous as it can harmlessly penetrate biological tissues and other non-magnetic materials[25], allowing for untethered operation[26–28]. Consequently, bio-inspired magnetic soft robots have attracted increasing attention, especially those embedded with ferromagnetic particles. These materials facilitate programmable, three-dimensional deformations in

small-scale soft robots[29], enhancing their functionality in tasks requiring precise manipulation, effective locomotion[12,28], pumping and valving[30–32], or greater actuation forces[2,27]. While progress has been made in exploring shape transformations[11,33,34] and locomotion mechanisms[12,35,36] of magnetically actuated soft robots, emulating the complex expansion and compression processes prevalent in natural biological systems, especially in thin-walled hollow structures, remains a challenge due to manufacturing difficulties.

The functionality of magnetic soft machines is achieved through structural design and magnetization distribution. The primary manufacturing methods include molding and casting, combined fabrication and assembly, and additive manufacturing (AM). Molding and casting are suitable for producing machines with simple geometries[2,12,28,37,38]. Combined fabrication and assembly allow for the

[1]State Key Laboratory of Robotics and Systems, Harbin Institute of Technology, Harbin 150080, China. [2]These authors contributed equally: Xianghe Meng, Shishi Li. ✉e-mail: xiehui@hit.edu.cn

creation of machines with specific three-dimensional shapes and magnetization distributions, though the increasing design complexity adds to the manufacturing difficulty and labor-consuming demands[1,33,39]. Additive manufacturing, advances the field by enables the production of robots with complex shapes and programmable magnetization distributions[34,40–42]. However, several challenges persist in printing soft machines that mimic biological functionalities by this method, particularly when imitating the thin-walled structures of organs with hollow and curved features and their dynamic behaviors.

One of the most critical challenges in printing thin-walled soft structures is their susceptibility to deformation or collapse. Different additive manufacturing approaches face unique challenges in addressing this issue. In direct ink writing (DIW) methods, gravity exacerbates the problem by acting on the already-formed structures, necessitating the addition of support materials[34,43]. Moreover, DIW typically constrains magnetization to the printing direction, limiting the achievable 3D magnetic patterns and reducing design flexibility. While the inclusion of heterogeneous support materials helps maintain structural integrity, it also increases printing difficulty and significantly reduces efficiency[44]. On the other hand, bottom-up digital light processing (DLP)-based patterned magnetization approaches[42,45] offer more freedom in creating complex magnetization patterns. However, these methods face their own set of challenges. Previously printed thin-walled structures are particularly prone to deformation due to orienting magnetic fields during the manufacturing process, which can lead to fabrication failures. As a result, both DIW and DLP methods are generally more suitable for planar structures or 3D soft robots with simple vertical wall structures. Additionally, these additive manufacturing processes for magnetic soft machines face common challenges such as low efficiency and difficulty in directly manufacturing intricate three-dimensional structures with integrated designs. The transition from 2D to complex 3D thin-walled structures with programmable magnetization remains a significant challenge in the field.

To address these challenges, we introduce a uniform magnetic field-assisted stereolithography (UMA-SL) method for fabricating 3D thin-walled biomimetic soft machines. This technique utilizes coaxially aligned Halbach arrays and an electromagnetic solenoid to generate a high-intensity uniform magnetic field (inhomogeneity of < 5 %, angle error <1. 5°) across the entire photocuring plane with low power consumption. This field precisely orients pre-magnetized magnetic particles, followed by DLP projection exposure to cure the pattern and achieve magnetic programming. To enhance the dispersion stability, printability, and curing quality of the magnetic soft materials, we optimized the composition of the ferromagnetic particle-filled resin. Additionally, a support addition strategy was developed, which involves defining support regions, adjusting support spacing, and determining support dimensions. This strategy maintains the intended shape and deformation capability while minimizing interference from magnetic fields and the influence of residual supports. The UMA-SL method facilitates efficient parallel processing, allowing the fabrication of magnetic soft machines with wall thicknesses as thin as 200 $\mu$m and hollow ratios up to 0.92, as well as programmable shape morphing. These advancements have the potential to enhance the functionality of soft machines in biomedical applications, including simulating cyclic fluid propulsion, unidirectional fluid transport, peristaltic deformation, active targeted cargo transport and delivery, and liquid biopsy collection.

## Results

### Uniform magnetic-assisted stereolithography
The UMA-SL system enables the fabrication of 3D structures with diverse magnetic orientations through a bottom-up photopolymerization technique (Fig. 1a). A magnetic field programs the orientation of magnetic particles, while UV light solidifies the material to preserve these arrangements.

The detailed process for fabricating soft magnetic materials with programmable magnetization is illustrated in Fig. 1c. This process involves integrating NdFeB particles into a photocurable resin matrix (Fig. 1b i). In the presence of a magnetic field, the NdFeB particles align with the field direction and are then solidified by a DLP projector in precise patterns (Fig. 1b ii). The entire fabrication occurs within a 40 mm diameter area under a uniform magnetic field, enabling the creation of intricate thin-walled magnetic 3D structures with varied magnetic distributions. After printing, the integrated magnetic structures and supports are easily removed, resulting in magnetic-driven soft machines.

At the heart of the UMA-SL system is an adjustable 3D magnetic field generator designed to ensure uniformity across the DLP projection plane ($\phi$40 mm) (Fig. 1a). Addressing the challenge of aligning ferromagnetic particles in a viscous UV-curable resin, this generator combines Halbach arrays with an electromagnetic solenoid to produce high-intensity magnetic fields up to 80 mT (see Supplementary Fig. 1), significantly reducing overall volume and energy consumption. A hollow solenoid allows light passage while uniformly generating a magnetic field along the z-axis. Three coaxially aligned Halbach arrays surround the solenoid, generating a uniform magnetic field at the curing plane while minimizing rotational inertia and volume. This setup enables adjustable magnetic fields, finely tuned by the angular displacement $\alpha$ between the arrays (see Supplementary Discussion 1), enhancing energy efficiency with ~23 W consumption using Halbach arrays and around 280 W with the solenoid.

Evaluation of the magnetic field's strength, direction, and uniformity within the curing plane was performed (Fig. 1d, e). Utilizing Halbach arrays and electromagnetic solenoids, the system generated an adjustable magnetic field, defined by parameters ($M$, $\theta$, $\varphi$), with $M \in$ [0 mT, 80 mT], $\theta \in$ [0°, 180°], and $\varphi \in$ [0°, 360°]. Precision measurements at the plane's center, with $M$ at 80 mT, showed field strength and directional errors within [0. 1°, 1. 3°] (Fig. 1d) and [78.7mT, 81.0 mT] (Supplementary Discussion 2 and Supplementary Fig. 2a). Simulations and measurements using a three-dimensional magnetometer within a 40 mm diameter area, set at (80 mT, 90°, 0°), indicated field strength inhomogeneity within 5% and a maximum directional error of about 1° (Fig. 1e)(see Supplementary Discussion 3). These results were consistent in various directions (Supplementary Discussion 2 and Supplementary Fig. 2b), confirming the system's effectiveness.

The magnetic moment density of printed samples was assessed under different conditions to evaluate their magnetic performance. Under a high-intensity originating magnetic field of 80 mT, the magnetic moment density increased almost linearly from 7.7 kA m$^{-1}$ – 16.6 kA m$^{-1}$ as the NdFeB content increased from 10% – 20%. This density is used in finite element simulations to predict the deformation of magnetic structures under external control fields. Samples magnetized under an 80 mT originating field reached 88%–90% of the maximum achievable magnetization, compared to those magnetized to saturation under a 5 T pulsed field (Fig. 1f).

### Magnetic patterning and performance of printed material
Manufacturing magnetic programmable 3D structures relies on the stability, printability, and curing quality of magnetic soft materials. The printing process, utilizing an orientating magnetic field and DLP technology, is influenced by the concentration of NdFeB particles and external magnetic fields. Dispersion stability is influenced by the size of NdFeB microparticles and the addition of silica nanoparticles, while curing depth and mechanical strength depend on NdFeB content and UV exposure.

Stable dispersion of magnetized particles during extended printing is maintained by incorporating rheological modifiers such as fumed silica nanoparticles. These nanoparticles increase the material's viscosity, improving dispersion stability and reducing particle

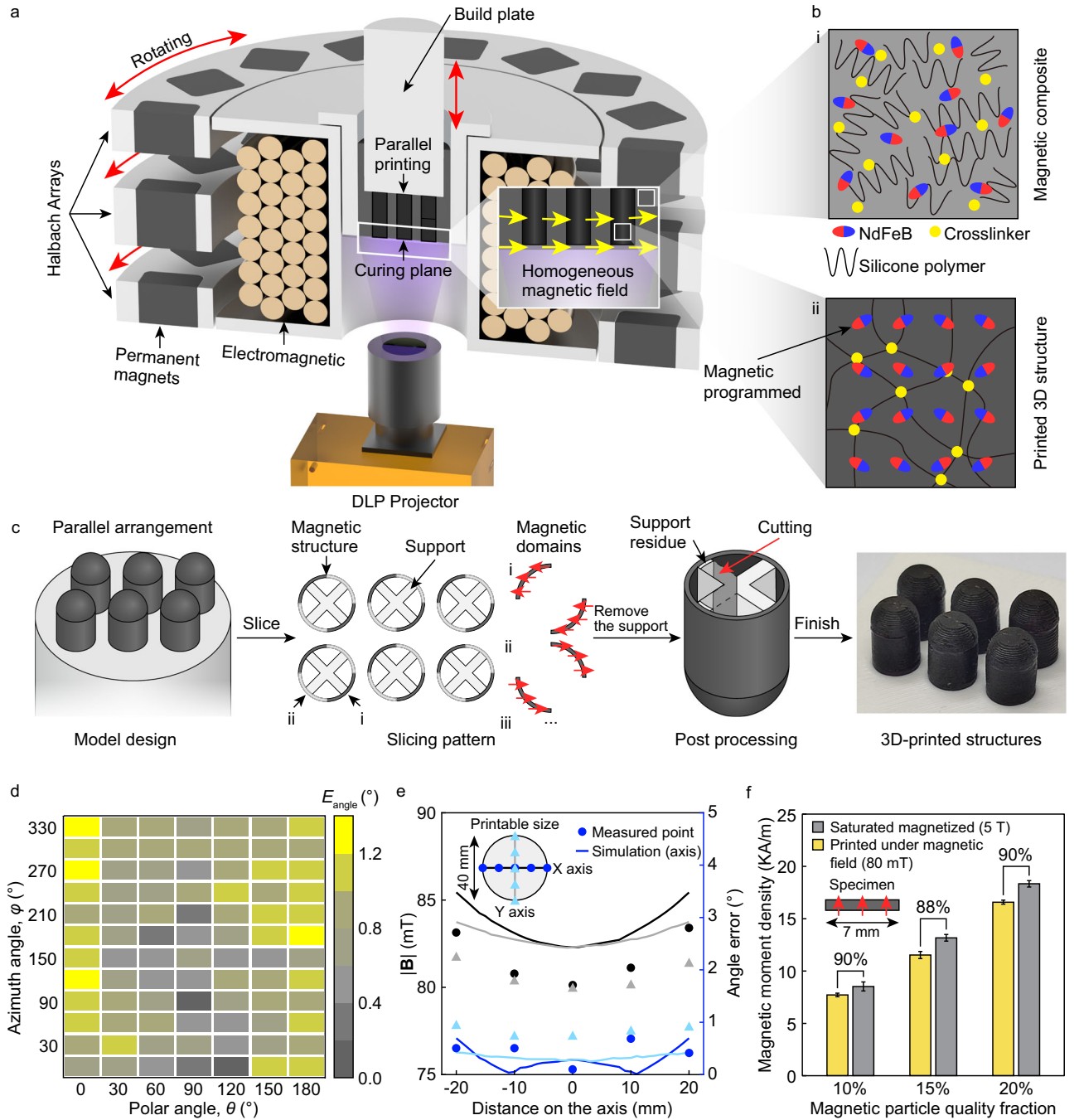

**Fig. 1 | Scheme of arbitrary 3D uniform magnetic field-assisted stereolithography method. a** Illustration of the printing principle showing the magnetic field generation apparatus that creates a target magnetic field to induce alignment of ferromagnetic particles within the printing material, with the DLP projector curing the material in the desired pattern, locking the magnetic particles in their aligned state. **b** Material composition of the uncured printing material comprising NdFeB particles, silicone polymer monomer, and a photoinitiator, with the cured material having crosslinked polymer monomer embedding the magnetic particles in the soft material matrix, enabling complex magnetic pattern programming. **c** Printing process involving slicing the model, simultaneously printing both the magnetic structure and support, then removing the support to obtain the final structure, with an arrayed model design significantly reducing printing time. **d** Accuracy of the magnetic field direction generated by the apparatus, which produces a uniform magnetic field of 80 mT in any arbitrary 3D direction, measured at the center of the printing plane. **e** Uniformity and accuracy of the magnetic field strength and direction evaluated within a 40 mm diameter area on the printing plane. **f** Magnetic moment density of printed samples with different NdFeB mass fractions, where the induced magnetic moment density (yellow bars) reaches 88%–90% of the saturation magnetization (gray bars). Error bars indicate the standard deviation for $n = 3$ samples at each data point. The inset shows the dimensions of the measured specimen (7 mm).

agglomeration. However, higher viscosity also creates greater resistance for particle reorientation, leading to lower residual magnetization and complicating resin flow in stereolithography. Achieving long-term stable dispersion of magnetic particles requires balancing viscosity and reducing the size of NdFeB particles (Supplementary Fig. 3), which is crucial for reliable 3D printing.

Four printing materials with varying NdFeB content were prepared: G1 (10 wt% ground 1–2 μm NdFeB + 2 wt% fumed silica), G2

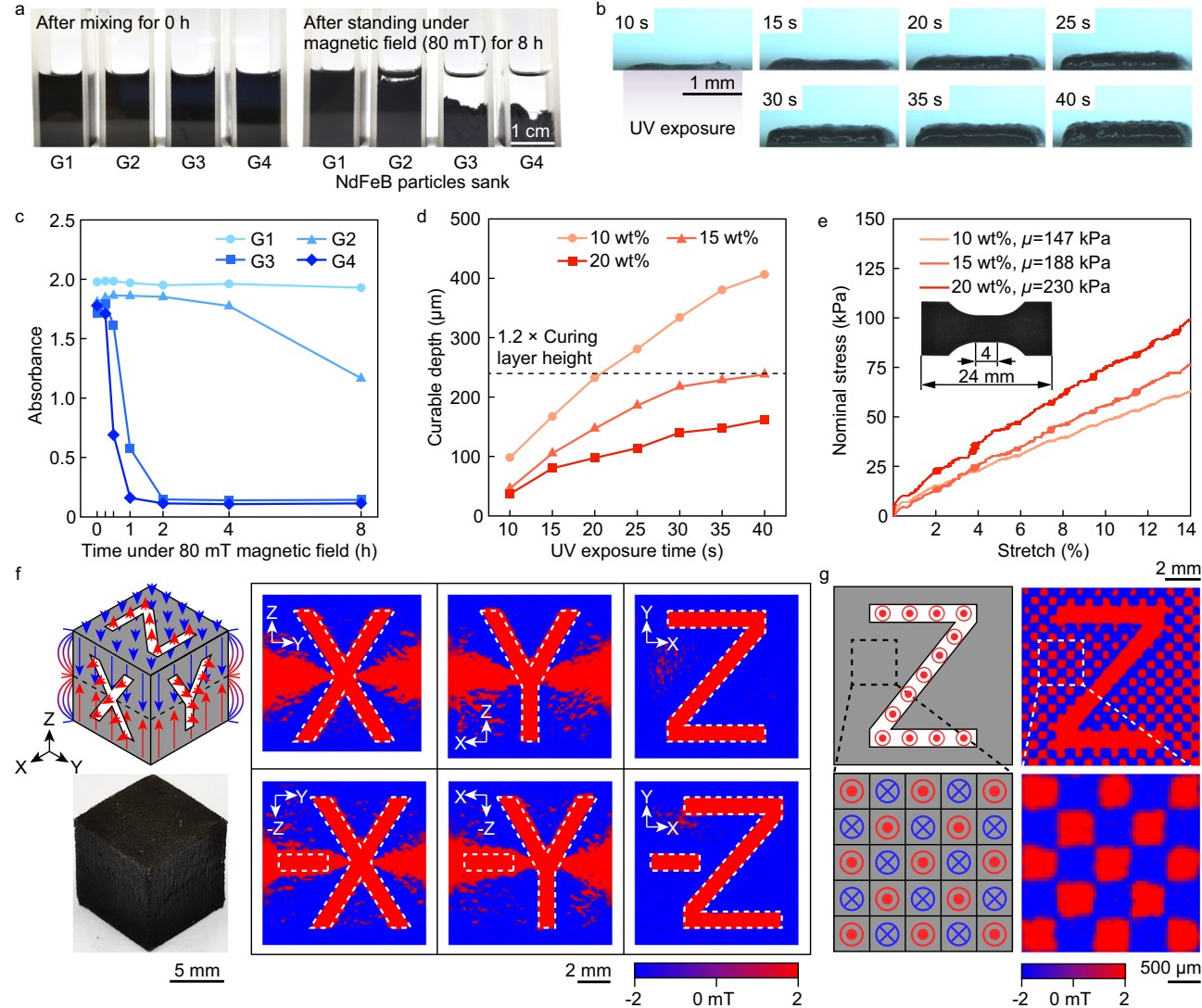

**Fig. 2 | Characterization of printing performance and 3D magnetic patterns.**
**a** Sedimentation of 10 wt% NdFeB printed material with different dispersion compositions (G1, G2, G3, G4) under an external magnetic field of 80 mT over time.
**b** Microscope images of the sidewall curing depth of 10 wt% NdFeB printed material at different UV exposure times. **c** Absorbance changes over time for 10 wt% NdFeB printed material with different dispersion compositions (G1, G2, G3, G4) under an external magnetic field of 80 mT. **d** Effect of UV exposure time on the curing depth of 10 wt% to 20 wt% NdFeB printed material. **e** Stress-stretch curves for tensile

samples printed with 10 wt% – 20 wt% NdFeB printed material. **f** Programming magnetic patterns and orientations on six faces of a cube. The magnetic patterns are visualized using a magneto-optical sensor, displaying different letters through varying magnetization directions. **g** Programming the magnetic orientation on one face to achieve zero net magnetic moment. The top row shows a programmed "Z" pattern, and the bottom row shows a checkerboard pattern. Imaging is done using a magnetic field imaging instrument (MagView S with type A sensor, Matesy GmbH).

(10 wt% commercial $5\,\mu m$ NdFeB + 2 wt% fumed silica), G3 (10 wt% ground $1–2\,\mu m$ NdFeB), and G4 (10 wt% commercial $5\,\mu m$ NdFeB). These materials were exposed to an 80 mT magnetic field, and their absorbance changes were monitored over time. Detailed information on the composition, proportions, and biocompatibility of these materials can be found in Methods. As shown in Fig. 2a, after 8 h, G1 remained stable, whereas G2, G3, and G4 exhibited significant sedimentation (Supplementary Fig. 4 and Supplementary Discussion 4). Absorbance curves (Fig. 2c) revealed that G4 agglomerated quickly, G3 remained stable for 15 min, G2 for 4 h, while G1's absorbance decreased by only 2% after 8 h. This indicates that smaller particles and fumed silica enhance stability. Thus, G1's composition will be utilized in future experiments.

The appropriate layer thickness and exposure time were determined by investigating the curing depth for exposure times ranging from 10s – 40 s with materials containing 10 wt%, 15 wt%, and 20 wt% NdFeB (Fig. 2b and Supplementary Fig. 5). Curing depth increased

linearly with the logarithm of exposure time, slowing after 40 s (Fig. 2d and Supplementary Discussion 5), and decreased almost linearly with increasing NdFeB content due to light absorption. To ensure good interlayer crosslinking, layer thicknesses of $200\,\mu m$ for 10 wt% and 15 wt%, and $100\,\mu m$ for 20 wt% were used, with exposure times set to 1.2 times the corresponding layer thickness curing time (see Supplementary discussion 6 for layer uniformity analysis). The Young's modulus increased from 421 kPa – 658 kPa, and the shear modulus from 147 kPa – 230 kPa with higher NdFeB content (Fig. 2e). The decreased flexibility may be attributed to magnetic particles acting as defects, as well as the longer exposure times required for higher NdFeB content.

The method enables directional programming of magnetization in 3D structures containing magnetized particles. A 10 mm cube was fabricated with unique magnetization patterns on each face, integrating components along the $x$, $y$, and $z$ axes, resulting in a comprehensive 3D magnetization distribution (Fig. 2f and Supplementary

Fig. 6). Invisible characters became visible using a magneto-optical sensor, matching the original design (Fig. 2f, top left). This demonstrates the method's capability in designing and programming magnetization distributions for individual layers. Creating structures with zero magnetic moment, such as for support, was achieved by proposing a pixelation method. Since magnetized particles in printed materials remain aligned with the previous stronger orienting field when the external field strength decreases to zero, achieving zero magnetic moment by simply reducing the field strength is not feasible. By alternating magnetization directions in adjacent pixelated areas, the entire printed structure achieves a net zero magnetic moment. Observations with a magneto-optical camera revealed regular alternating magnetization directions and a clear "Z" pattern (Fig. 2g), confirming the method's effectiveness.

## Supports generation strategy

Orienting magnetic fields and shear forces during the 3D printing process can cause deformation in magnetic soft structures, compromising their quality and success rates. To counteract this, an optimized strategy places and sizes supports specifically for thin-walled structures. This method ensures the structures maintain their intended shape and deformation capability, free from negative impacts during the printing process or residual supports.

Finite element method (FEM) simulations analyzed the strategy's effectiveness on a tubular thin-walled magnetic 3D structure with a 10 mm diameter and 300 μm wall thickness. Fig. 3a shows the magnetization direction distribution. Subjected to an external magnetic field, the structure bends, changing its cross-sectional shape from a circular ring to a compressed elliptical shape. The FEM simulations identified four strain concentration points and two maximum deformation points.

Evaluations examined three support addition regions: unconstrained, avoiding maximum deformation points, and avoiding strain concentration points (Fig. 3b). Supported structures had 500 μm thick residual supports, while the ideal structure had none. Comparing volume contraction ratios revealed that supports avoiding maximum deformation points reduce the volume contraction ratio by 92%, significantly weakening the deformation ability. Conversely, supports avoiding strain concentration points increase the volume contraction ratio by 21%. Strategically placing supports mitigates negative impacts on deformation and enhances the deformation capacity of magnetic thin-walled structures. This improvement results from an increased magnetic moment in the supported regions, creating a deformation gain effect. To preserve the desired deformation properties, it is essential to avoid placing supports at strain concentration points.

Figure 3c illustrates the influence of support spacing on the ability of magnetic thin-walled structures to resist disturbance deformation during the printing process. Simplifying the structure between supports as a beam with fixed ends helps quantify this effect, given the minimal deformation of supports and the fixed constraint they provide. Two magnetization patterns, opposite magnetization directions and a single magnetization direction, were analyzed. An external magnetic field of 80 mT was applied perpendicular to the magnetization direction to simulate maximum disturbance during printing. Simulations (Supplementary Fig. 7a, b) analyzed four wall thicknesses (200 μm, 300 μm, 400 μm, 500 μm) and support spacings of 1–5 mm (with 1 mm intervals). For opposite magnetization, deformation increased with spacing and decreased with thickness. Single magnetization resulted in nearly zero deformation. To ensure interlayer bonding and avoid defects, disturbance deformation was limited to 10% of the wall thickness, guiding support spacing selection.

The effect of support size on resisting disturbance deformation was also assessed. Supports modeled as cantilever beams, with one end connected to other supports and the other to the printed structure, employed pixelation to achieve zero overall residual magnetization. The support-structure connection was 500 μm thick with the same magnetization direction as the connected structure. Simulations analyzed support widths of 0.5 mm, 1.0 mm, and 1.5 mm, with lengths ranging from 1–5 mm (with 1 mm intervals) for a width of 0.5 mm and 2–10 mm (with 2 mm intervals) for widths of 1.0 mm and 1.5 mm under an 80 mT field (Supplementary Fig. 7c). Fig. 3d shows deformation increased with support length and decreased with increased support width. Support deformation was limited to 100 μm to prevent print failure, and appropriate shapes were selected based on available space.

Applying this method, a tubular 3D thin-walled magnetic structure with a hollow ratio of 0.92 (200 μm wall thickness, 10 mm diameter, and 10 mm height) was successfully printed (Fig. 3e). Support addition regions avoided the four strain concentration points after deformation. The support spacing was 1.5 mm, and the support shape was 2.8 mm long and 1 mm wide, meeting the strategy requirements. Residual supports shorter than 500 μm were observed after removal (Fig. 3e, right). Under a 30 mT control magnetic field, the structure folded inward, compressing the internal cavity, consistent with simulations.

## 3D structures with programmable magnetization

By leveraging the developed support strategies and precise magnetic programming capabilities, UMA-SL printed a series of 3D thin-walled structures with complex geometries and diverse magnetic properties. These structures exhibited distinct deformations and functionalities under external magnetic fields, enabling the realization of key design features that were previously challenging to achieve, including curved forms, hollow thin-walled constructions, and array-based multi-unit structures.

A 3D magnetic check valve structure was designed and fabricated, mimicking the shape and function of the human heart's arterial valve (Fig. 4a). This structure consists of a valve annulus, diameter: 8 mm, wall thickness: 800 μm, and three valve leaflets, free edge length: 6.4 mm, wall thickness: 400 μm, with large-span curved surfaces, each spanning a 120° arc. The annulus has zero magnetic moment and provides physical constraints for the leaflets. The leaflets are magnetically oriented towards their tips. (For details on the magnetization profile design, see Methods.) When exposed to a specific magnetic field direction, the leaflets bend inward, resulting in a triangular orifice morphology that enables forward flow conduction. Under a reverse magnetic field, the leaflets open outward, with their free edges constraining each other. This results in a closed orifice morphology, achieving reverse flow closure.

Furthermore, a biomimetic study of the human heart's mitral valve led to the design and fabrication of a 3D magnetic check valve structure (Fig. 4b). This structure includes a valve annulus (diameter: 13 mm, wall thickness: 1 mm) and anterior and posterior leaflets (edge length: 11 mm, wall thickness: 500 μm) with large-span curved surfaces and intersecting free edges. The magnetic distribution mimics the pulling effect of the chordae tendineae and papillary muscles. When exposed to a specific magnetic field direction, the leaflets fold outward, creating an elliptical orifice morphology. A reverse magnetic field causes the leaflets to fold inward and tightly adhere, resulting in a closed orifice, thus mimicking the forward flow conduction and reverse flow closure functions of the human mitral valve. These magnetic valve structures enable unidirectional fluid flow control, achieving non-contact, field-controlled check valve functionality.

To showcase the complex magnetic pattern programming capabilities of the proposed method, a thin-walled structure consisting of two adjacent octagonal tubes (wall thickness: 500 μm) was fabricated (Fig. 4c). The magnetization is designed to be perpendicular to the curing plane. The opposing magnetization directions in upper and lower parts of each tube and between adjacent tubes result in alternating concave and convex deformations, creating an undulating surface. Figure 4d presents a multi-unit honeycomb structure, where each

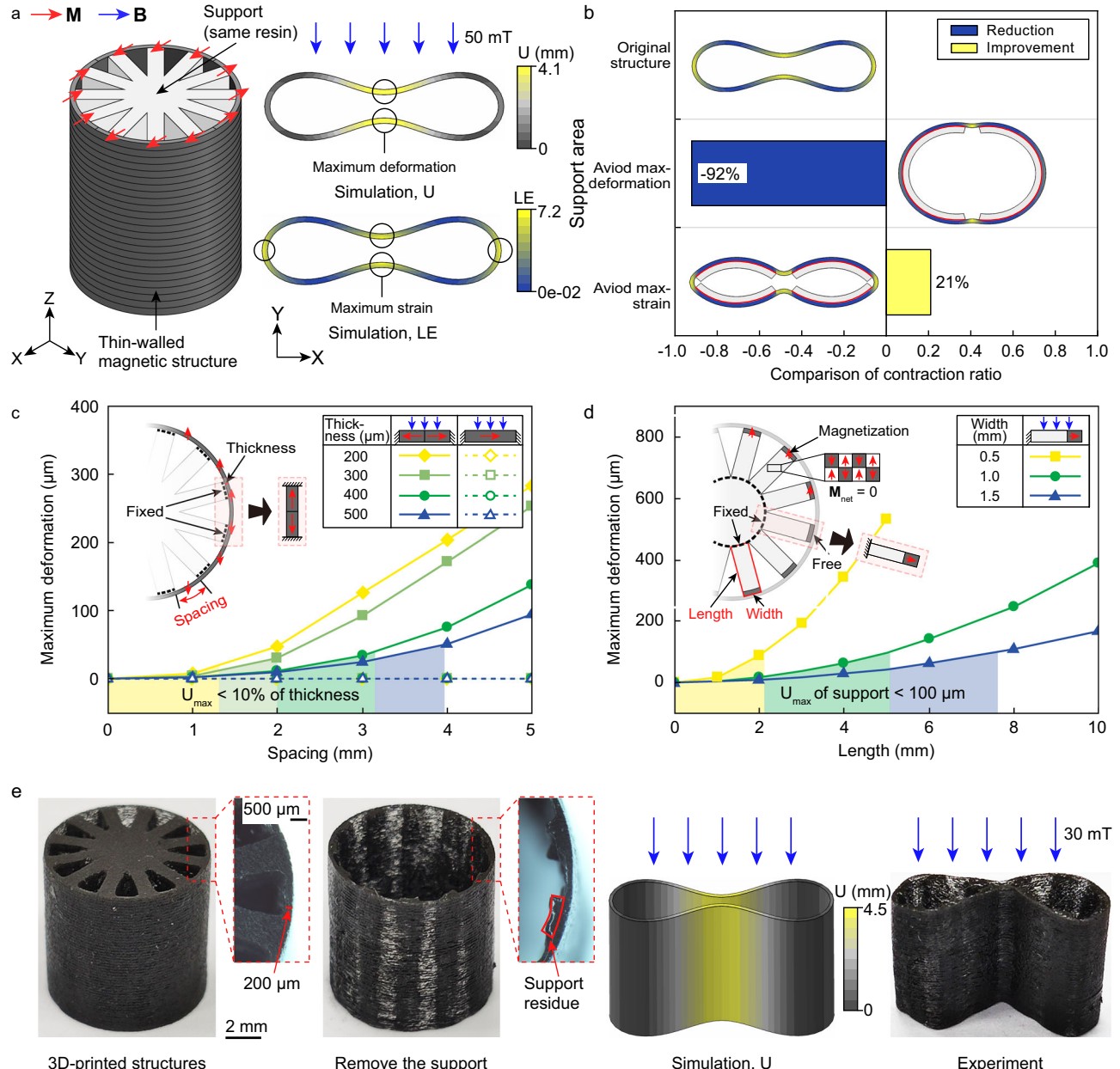

**Fig. 3 | Strategy for printing thin-walled magnetic structures. a** Schematic of the 3D design of a tubular magnetic thin-walled structure and finite element simulations of deformation and strain distribution under an external magnetic field of 50 mT. **b** The effect of adding support regions on the deformation capacity of the magnetic thin-walled structure, comparing maximum deformation and strain distribution before and after adding support regions. **c** The effect of support spacing on the resistance to deformation disturbances during the printing process of thin-walled magnetic structures, showing the relationship between maximum deformation and support spacing for different wall thicknesses. **d** The effect of support width on the resistance of the supports themselves to deformation disturbances during the printing process, showing the relationship between maximum deformation and support length for different widths. **e** The printed results of a tubular thin-walled structure with a wall thickness of 200 $\mu m$, images of the structure after support removal, and comparisons of deformation finite element simulations and experimental results under an external magnetic field of 30 mT.

hexagonal unit is divided into 4 magnetization zones with magnetization directions at angles to the printed structure edges. When exposed to an external magnetic field, the structure undergoes uniaxial tensile deformation. Each unit exhibits consistent behavior and magnitude. This demonstrates the uniformity of the printed material composition, induced magnetic field, and exposure intensity within the large-scale printing range, confirming the reliability of efficient parallel printing.

**Biomimetic colonic peristaltic machine**

A magnetic 3D extrusion structure designed to mimic intestinal peristaltic fluid transport was developed and fabricated. The structure, designed to be 40 mm in length, 10 mm in diameter, and with a wall thickness of 500 $\mu m$, features a magnetization distribution that divides its cross-section into four distinct regions. Under the influence of an external driving magnetic field, portions of the tubular structure undergo compressive deformation, transforming the cross-sectional shape from circular to elliptical, effectively simulating intestinal contraction (Fig. 5a). To enhance manufacturing efficiency, the structure comprises four bonded 10 mm-long segments, each with positioning points for easy assembly, and is produced using parallel printing techniques. The design and assembly details of the machine are presented in Supplementary Fig. 9.

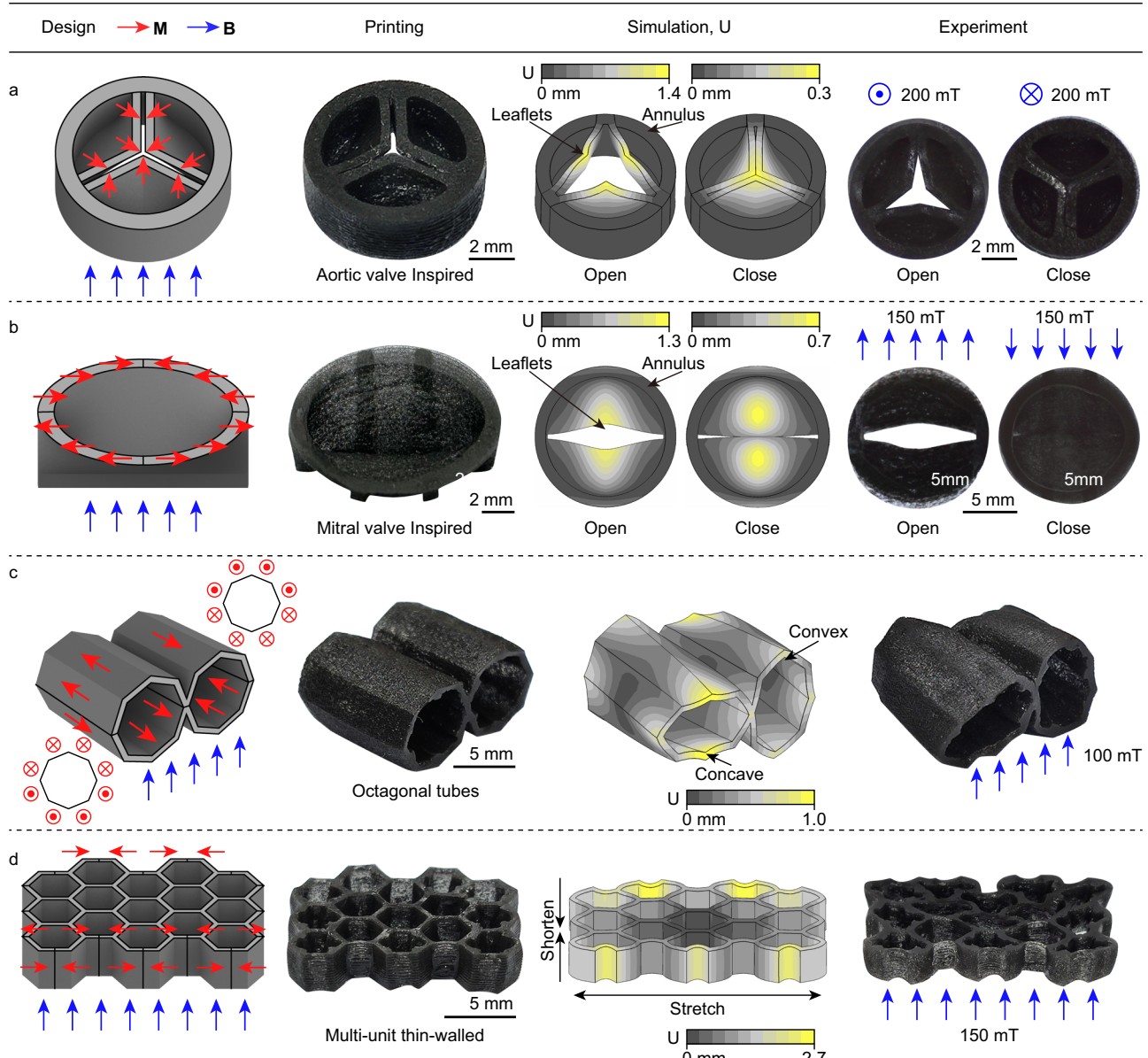

**Fig. 4 | 3D printed magnetic soft structures with programmable deformations.** **a-b** Two types of one-way valves inspired by aortic and mitral valves, which can open and close under alternating magnetic fields (**a**: 200 mT, **b**: 150 mT). **c** Two adjacent octagonal tube structures with magnetization perpendicular to the curing plane, demonstrating alternating concave and convex deformations. **d** A multi-unit thin-walled structure composed of thin-walled hexagonal units, showing uniaxial elongation along the field direction under a 150 mT magnetic field. All structures were fabricated using a magnetic soft material containing 10 wt% magnetized NdFeB particles. Detailed dimensions, strain distribution simulation results, and supports are provided in Supplementary Fig. 8.

The effect of programmable magnetic distribution on the structure's extrusion performance was investigated (Fig. 5b). The magnetic torque on the structure in a uniform magnetic field is given by $T_m = V_m|\mathbf{M} \times \mathbf{B}| = V_m|\mathbf{M}||\mathbf{B}|\sin\theta_0$, where $V_m$ is the volume of the structure and $\theta_0$ is the angle between the magnetization direction and the applied control magnetic field. The magnetic torque reaches its maximum when $\theta_0 = 90°$. During deformation, $\theta_0$ decreases, aligning the magnetization direction with the applied field, and the magnetic torque gradually equilibrates with the stress generated by elastic deformation. To determine the optimal initial angle for maximum extrusion capability, extrusion structures with magnetic distributions of $\theta_0 \in \{90°, 105°, 120°, 135°\}$ were fabricated and evaluated through simulation and experimental studies. As $\theta_0$ increased from 90° – 135°, the simulated volume contraction ratio rose from 0.44 to 0.66, while the experimental ratio increased from 0.57 – 0.68. The optimal initial

angle between the magnetization direction and the external magnetic field for maximum extrusion capability was found to be 135°. Beyond this angle, the volume contraction ratio approached a constant value due to the physical contact of the inner wall surfaces, which limited the maximum deformation capacity. The difference between simulated and experimental volume contraction ratios decreased as the deformation capacity reached its limit, attributable to the driving force enhancement effect of residual supports present in the experimental structures.

The extrusion machine can directionally transport viscous fluids, with the transport direction governed by the motion of the applied driving magnetic field (Fig. 5c, Supplementary Movie 1). A permanent magnet generating a 150 mT field with an effective width of 10 mm is used. The field is initially applied at the distal end, inducing compressive deformation and increasing the pressure of the internal

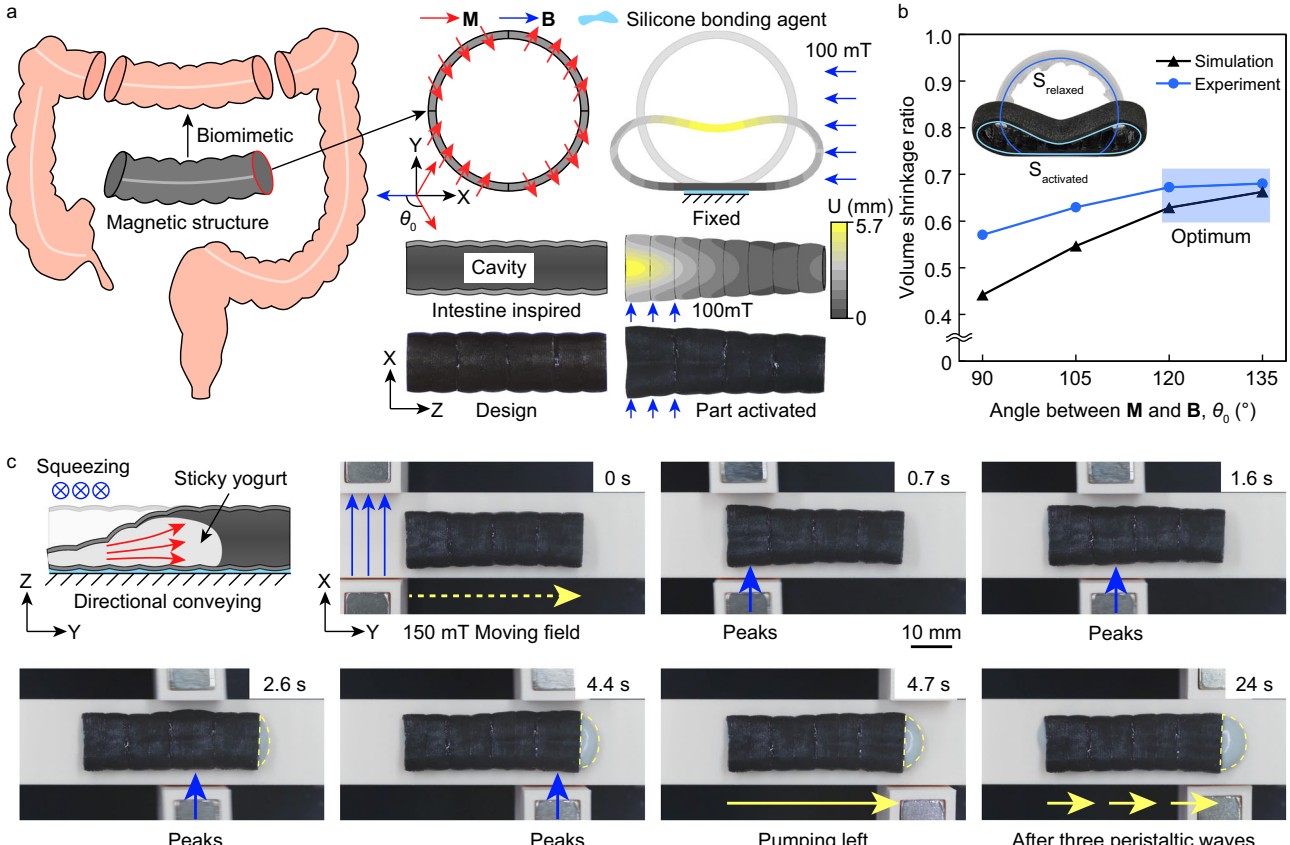

**Fig. 5 | Magnetic 3D extrusion machine mimicking the human intestinal tract.**
**a** A 3D magnetic extrusion machine inspired by the colon, exhibiting partial tubular deformation under the influence of a localized driving magnetic field. **b** Extrusion performance of the soft peristaltic machine at different initial magnetization angles ($\theta_0$), with simulation results (black triangles) compared to experimental results (blue circles). **c** Image sequence showing forward-propagating deformation waves, peak positions, and directional conveying path at different time points under a moving driving field. Videos of all tests can be found in Supplementary Movie 1.

viscous fluid (yogurt mixed with 2% fumed silica nanoparticles), causing it to propagate forward. As the driving field progresses, the deformation region advances, forming a peristaltic wave that propels the fluid to the proximal end until extrusion, mimicking the intestine's peristaltic defecation function.

## Biomimetic heart pumping machine

Inspired by the structural and functional characteristics of human heart ventricles, a magnetic three-dimensional pumping structure was designed to emulate their behavior. The shape of this pumping structure is simplified to resemble a half cone with a diameter and height of 24 mm. Its interior is hollow for fluid storage, with a wall thickness of 600 $\mu$m. The pumping mechanism operates by mimicking the diastolic and systolic motions of the ventricles. During the diastolic phase, the internal cavity volume of the pumping structure increases, allowing it to collect inflowing fluid. Conversely, during the systolic phase, the structure compresses the internal cavity, causing the internal pressure to rise rapidly and pumping out the stored fluid (Fig. 6a). An alternating magnetic field enables the pumping structure to continuously expand and contract, thus mimicking the function of human ventricles in pumping blood.

The ventricle-mimicking pumping system includes a pumping structure as the driving source, a fluid inlet and outlet, and a magnetic unidirectional valve that controls flow direction (Fig. 6b). The assembly process involves removing support from the printed ventricle-mimicking inner wall, attaching a zero magnetic moment thin plate to the opening with silicone to form a closed cavity (design details in Supplementary Fig. 10), and installing two unidirectional valves on the inner wall of the silicone tube with opposite directions, connected to

the opening membrane, ensuring unidirectional fluid flow. The magnetic unidirectional valve, designed with zero magnetic moment, achieves unidirectional conduction through a structural design similar to the heart's pulmonary valve, positioned far from the driving field to avoid field reversal effects. The pumping structure's magnetization design divides it into two regions with symmetrical magnetization directions along the vertical central axis.

To investigate the influence of magnetic programming on pumping performance, the ejection fraction of pumping structures with various magnetic designs was studied (Fig. 6c and Supplementary Movie 2). The ejection fraction, the ratio of ejected fluid volume to end-diastolic volume, varied with different magnetization schemes, causing initial angles between the pumping structure's magnetization and the driving magnetic field to range from 90° to 135°. This led to opposing magnetic torques, folding the structure inward to compress the cavity. Under a 160 mT field, the 120° design achieved the highest ejection fraction, increasing 1.8-fold (from 10.9% at 90° – 19.7%) with a 1 s field application. Conversely, the 135° design had the lowest at 9.5% due to minimal deformation from high fluid resistance. Testing various field application times (1 s, 5 s, 10 s, 20 s), the 120° design consistently proved optimal, with ejection fractions rising 4.7-fold (from 19.7% to 93.0%), indicating near-complete fluid expulsion. Thus, the 120° magnetization design enables maximum pumping capacity in the biomimetic soft pump.

The operation of the pumping system is illustrated in Fig. 6d and Supplementary Movie 2. Under the influence of a leftward driving magnetic field, the ventricle-mimicking pump reaches the end-diastolic phase, during which the fluid pressure opens the inlet unidirectional valve channel, allowing the cavity to fill with liquid. When a

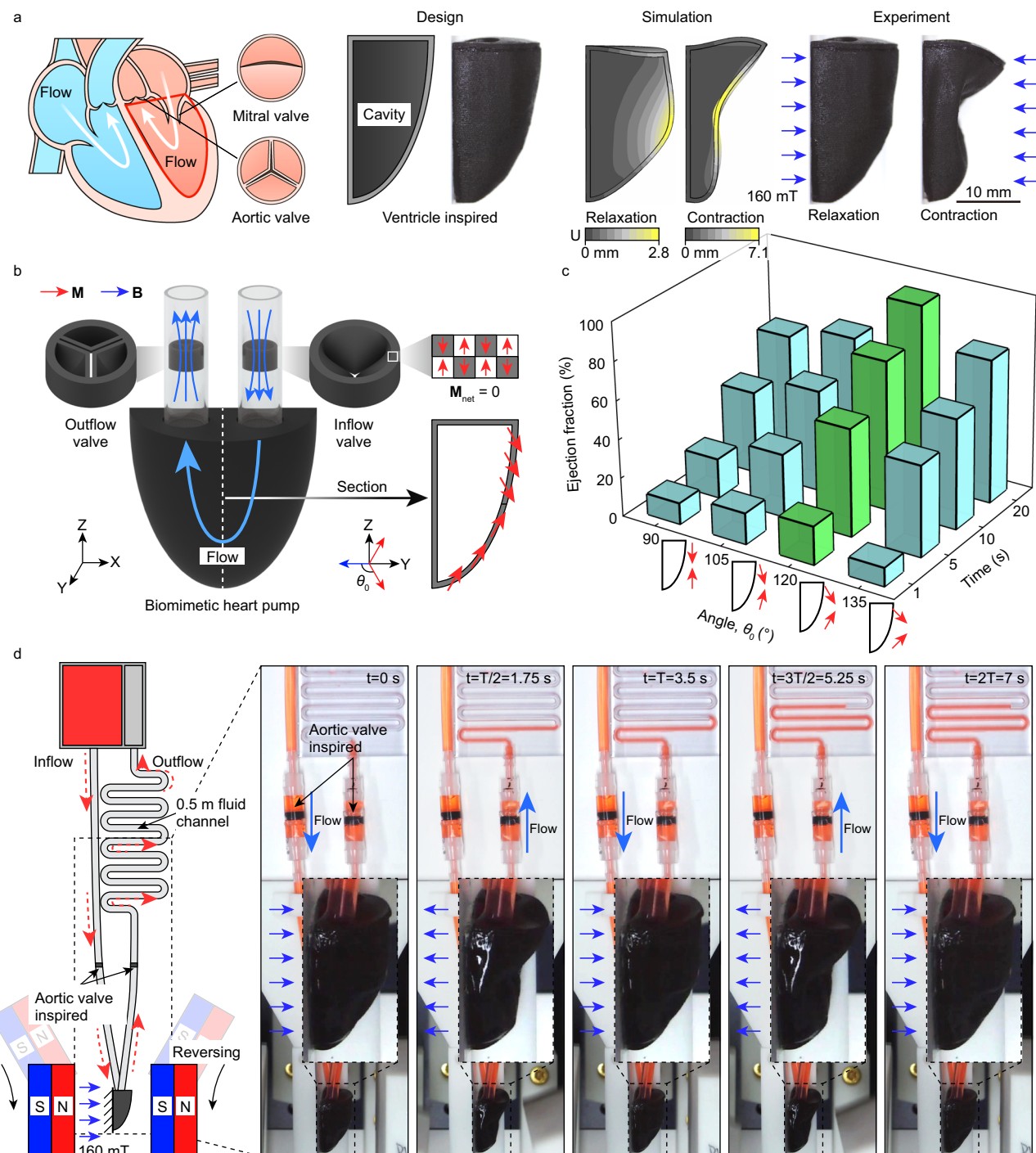

**Fig. 6 | Design and functionality of biomimetic soft pumping machine.**
**a** Ventricle-inspired 3D pumping machine mimicking the diastolic and systolic functions under alternating magnetic fields. The design, simulation, and experimental results show contraction and relaxation under a 160 mT magnetic field. **b** Structural design and magnetization distribution of the pumping model with magnetic unidirectional valves controlling fluid flow. The section view shows fluid flow direction and required magnetic field orientation. **c** Effect of programmable magnetic distribution on ejection fraction. Bar chart shows ejection fraction percentage for different angles and times. Optimal distribution is marked in green, others in blue. **d** Pumping experiment in an ex vivo setup mimicking a human blood network. Sequential images show fluid flow through the pump at different time points. Arrows indicate fluid direction and 160 mT magnetic field reversal.

reverse magnetic field is applied, the pump begins to contract, increasing the pressure inside the cavity. This increased pressure opens the outlet unidirectional valve channel and closes the inlet valve, forcing the fluid out of the cavity and into the simulated vascular network. As the magnetic field reverses again, the pump re-enters the diastolic phase, allowing fluid to flow in and refill the cavity, preparing for the next contraction cycle. The alternating magnetic field is generated by the periodic 180° rotation of two pairs of opposing permanent magnets (Supplementary Fig. 11). The vascular network is simulated using a channel with a diameter of 3 mm and a length of ~0.5 m, with the fluid being water mixed with red dye. Through the action of the alternating magnetic field, the pumping system successfully replicates cyclic fluid propulsion, mimicking biological pulsatile flow mechanisms. Detailed information about the fluid

pressure-controlled valve system and its coordination in the pumping process can be found in the Supplementary Discussion 7.

## Parallel-manufactured soft capsules with biomimetic intake-expulsion

A soft capsule robot capable of mimicking biological intake-expulsion behavior has been developed. These robots can carry liquid cargo, navigate to target locations by rolling, and perform precise drug delivery and sample collection. Each capsule has a diameter of 6 mm, a height of 11.2 mm, and an internal cavity for cargo, with a wall thickness of $500\,\mu m$. The structure consists of a cylindrical body and two hemispherical heads bonded with silicone rubber. A 150° arc-shaped slit with a width of $300\,\mu m$ is designed in the head for loading and releasing drugs. The magnetic design gives the capsule a net magnetic moment along the radial direction, allowing it to navigate under a weak rotating magnetic field. When a strong magnetic field is applied, the capsule orients and activates, causing the head slit to open and the internal cavity to compress, mimicking biological intake-expulsion for drug delivery or sample collection (Fig. 7a and Supplementary Fig. 12).

Using a highly precise and uniform induced magnetic field within the workspace, six capsules were manufactured in parallel. These capsules were deployed as a team, moving and activating in unison under external control, which increased the amount of cargo released at the target site (Fig. 7b). Future designs of heterogeneous robots (Supplementary Discussion 8 and Supplementary Movie 4) could enable the independent control of each capsule, allowing the team to release drugs at multiple target points as needed.

The deformation of the capsules under varying external magnetic field strengths was investigated (Fig. 7c) to determine the optimal control and activation field strengths, thereby avoiding cargo leakage due to deformation during transportation. Finite element simulations and experimental measurements showed that at 40 mT, the capsule's deformation was minimal, maintaining the cargo load until activation. As the magnetic field strength increased to an effective level (200 mT), significant deformation of the head slit and an increase in internal cavity pressure were observed, indicating system activation. It is also possible to adjust the required driving field through structural optimization, as detailed in Supplementary Discussion 9.

The loading, sealing, and release processes of the capsule are depicted in Supplementary Fig. 13 and Supplementary Movie 3. Initially, the capsule rolls into the loading chamber with red dye under a 30 mT and 0.1 Hz rotating field. It compresses the cavity and opens the head slit under a 200 mT magnetic field, expelling gas and ingesting the dye, mimicking biological feeding. This is repeated 2–3 times to ensure complete loading. The capsule then rolls uphill, crossing into the sealing chamber with paraffin oil. Rolling in paraffin oil coats the capsule with an oil film, creating a hydrophobic seal. Finally, the capsule rolls into the release chamber with water, releasing the cargo upon activation. This demonstrates the capsule's ability to perform loading and release tasks while maintaining cargo integrity with a hydrophobic coating until reaching the target area.

The capsule's cargo transport capabilities were also examined. Under a 30 mT and 0.2 Hz rotating field, the capsule was controlled to move along a hexagonal trajectory, maintaining its internal cargo for 10 min. Upon applying a 200 mT activation field, the capsule released the drug (Fig. 7d). The red dye inside the capsule showed no visible diffusion during rolling, indicating that the design could meet the needs of long-distance transport to hard-to-reach locations. Beyond on-demand drug delivery, the capsule can also be used for liquid biopsy tasks. By controlling the air-filled capsule to move into a target liquid, activating the it to expel air and collect a liquid sample (red dye), and then moving to a collection chamber to release the sample onto medical cotton (Fig. 7e).

Capsules can also operate as a team, moving and transporting drugs simultaneously. Under a 40 mT and 0.2 Hz rotating field, four capsules rolled across the uneven surfaces of a 1:1 human stomach model to the target area, where they simultaneously activated to release more cargo. After release, the capsules returned along the same path to exit the working area (Fig. 7f). These wireless magnetic soft capsules show significant potential for targeted drug delivery, liquid biopsy collection, and other therapeutic applications.

## Discussion

The uniform magnetic field-assisted stereolithography method presented in this work enables the printing of 3D thin-walled soft machines with hollow structures, curved surfaces, and programmable anisotropic magnetization. This technique enhances the design freedom of soft machines, allowing for complex deformations and motions. The UMA-SL method generates a high-intensity, homogeneous 3D magnetic field using Halbach arrays and an electromagnetic solenoid, enabling precise control of ferromagnetic particle orientation. This capability, combined with a tailored support strategy, results in thin-walled structures with high hollow ratios up to 0.92 and complex magnetic patterns.

The programmability of UMA-SL optimizes magnetic properties, resulting in significant improvements in various performance metrics of the fabricated soft machines. While currently suited for small-scale soft machines, this approach has the potential to be extended to larger printing ranges. Additionally, the soft, thin-walled nature of these machines allows for potential volume compression through curling or folding, which could be advantageous for certain applications. Preliminary cytotoxicity tests of PDMS-coated printed parts showed promising results (Supplementary Fig. 14), but comprehensive studies would be required for any biomedical considerations. Future work should focus on optimizing material properties, improving long-term stability of magnetic particles, and exploring a wider range of potential applications in soft robotics and related fields. Scaling up the printing process to create larger and more complex structures presents an exciting challenge. Addressing these aspects through interdisciplinary collaboration may further expand the capabilities of UMA-SL, potentially contributing to advancements in the design and fabrication of sophisticated soft machines for various applications.

## Methods

### Printing material composition and preparation

Two silicone methacrylate macromers were blended in a 1:1 weight ratio to create the elastomer component: (Methacryloxypropyl) methylsiloxane-dimethylsiloxane copolymer (PDMS-S) and Methacryloxypropyl terminated PDMS Fluids (PDMS-E) (both from Zhengzhou Gecko Scientific Inc.). PDMS-S has 3.5 times more reactive (methacryloxypropyl) groups than PDMS-E, so PDMS-E was used to reduce cross-linking, resulting in a softer elastomer.

Next, 1 wt% Ethyl(2,4,6-Trimethylbenzoyl)-phenyl phosphinate (TPO-L, Shanghai Curease Chemical Inc.) and 2 wt% fumed silica nanoparticles (FSN, 7–40 nm, Aladdin) were added to increase viscosity, forming a stable suspension. The mixture was blended with a planetary mixer (SK-300SII, Kakuhunter) at 2000 rpm for 1 min.

Then, $1–2\,\mu m$ NdFeB microparticles, 20 wt% of the composite, were added. They were first stirred for 2 min with a glass rod, then mixed at 2000 rpm for 2 min, and defoamed at 2200 rpm for 1 min. The NdFeB microparticles were magnetized under a −5 T impulse magnetic field (J8001, Nanjing Golden Wolf Electric Co.). These particles were prepared by milling $5\,\mu m$ NdFeB microparticles (MQFP-B-2007609-089, Magnequench) with a planetary ball mill (YD-YXQM-4L, Changsha Miqi Inc.).

### Printing procedure

The prepared printing material was loaded into a resin container mounted coaxially inside the cylindrical workspace of the three-dimensional magnetic field generation device, with the printing plane

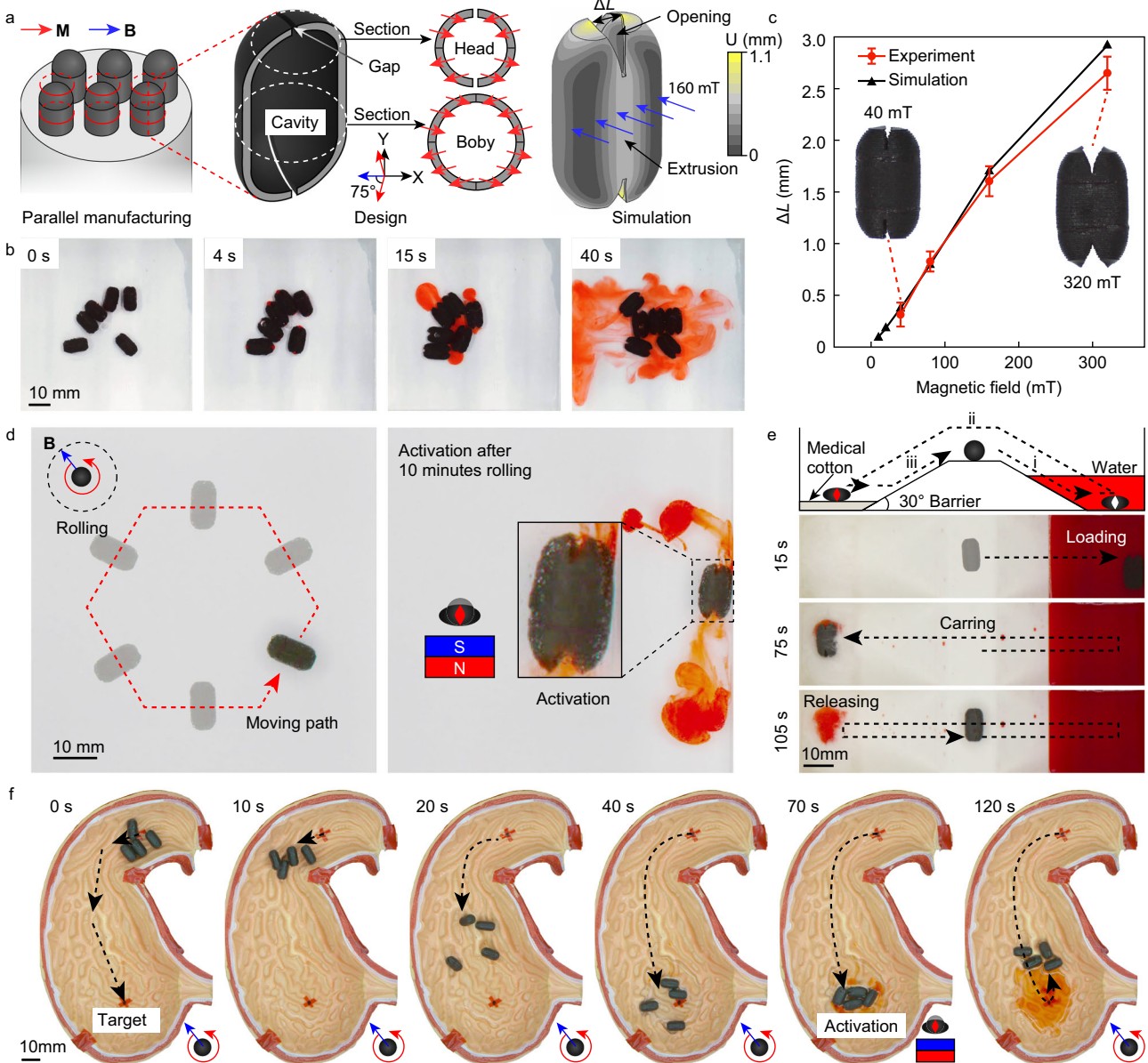

**Fig. 7 | Magnetic soft capsule robots with biomimetic intake-expulsion behavior and medical potential. a** Structural design, magnetization design, and activation state of the capsule. **b** Multiple capsules manufactured in parallel and simultaneously activated to release internal dummy drug liquids. **c** Relationship between deformation of the capsule and external magnetic field strength. Error bars indicate the standard deviation for $n = 3$ samples at each data point. **d** Movement of the capsule along a hexagonal trajectory under 30 mT and 0.2 Hz, maintaining cargo until release after 10 min. **e** Under 30 mT and 0.1 Hz, the capsule moves to the sample chamber to collect liquid samples, then moves to the collection chamber to release the samples onto medical cotton. **f** Under 40 mT and 0.2 Hz, the capsules move on uneven surfaces of a 1:1 human stomach model to the target area and activate to release the drug. The videos of all tests are available in Supplementary Movie 3.

aligned to its center plane. The printer head, resin container, and DLP projector (405 nm, DLP4500, Texas Instruments) were vertically aligned (Fig. 1a). The printer head moved vertically to print the structure layer by layer. The DLP projector adjusted its distance from the printing plane to ensure proper focus.

The printing model was created using three-dimensional modeling software (Rhino, Robert McNeel & Assoc) and sliced into layers using commercial slicing software (JuXinAdditive, Zhongshan Huayuyuanxin Inc.). A custom Python program divided each layer's pattern into regions based on the magnetic domain distribution and generated the exposure sequence (Fig. 1b).

The printing process was controlled through precise magnetic field parameters and exposure sequences, derived from the model's structure and magnetic distribution. Key steps included moving the

printer head, generating the magnetic field, and projecting the exposure sequence. Initially, the printer head pressed down until the gap between it and the printing plane equaled the layer thickness. The rotation angle of the Halbach arrays and the input current of the electromagnetic solenoid controlled the magnetic field, aligning the magnetic particles.

## Magnetization profile design process

The magnetization profiles for the demonstrated structures are designed using a function-based reverse design strategy, as illustrated in Supplementary Fig. 15. This methodology integrates physical principles with computational optimization to achieve desired deformations and motions. The process begins with the definition of a target deformed shape based on specific functional requirements. By

comparing the undeformed and target shapes, the required torque distribution is determined.

The initial magnetization profile is derived by equating the required torque to magnetic torque, expressed as $T_m = V_m|\mathbf{M}||\mathbf{B}|\sin\theta$, where $\theta$ represents the angle between magnetization and external field directions. This initial profile then undergoes optimization tailored to specific task requirements. For enhanced deformation, $\theta$ is adjusted to maximize torque during the deformation process, as demonstrated in Fig. 5b and Fig. 6c. To achieve controlled motion, $\theta$ is reduced to maintain a non-zero net magnetic moment along the field direction, as shown in Fig. 7c.

Simulation plays an integral role in the design process, serving multiple functions. It is first used to validate the efficacy of the initial magnetization profile in achieving desired deformations under applied magnetic fields. During the optimization phase, simulation helps quantify deformation magnitudes, guiding the refinement of magnetization distribution for maximum deformation, as illustrated in Fig. 5b and Fig. 6c. Additionally, simulation is crucial in determining appropriate locations for support structures. Logarithmic strain distribution maps of the deformed model are generated to identify strain concentration areas, defined as regions where strain exceeds 90% of the maximum strain. This information guides the strategic placement of support structures to minimize their impact on deformation, as demonstrated in Fig. 3b.

The methodology combines systematic reverse design with elements of physical intuition in the initial stages of defining target deformations and setting up the inverse design problem. These initial steps are followed by rigorous computational optimization and extensive use of simulation for validation and refinement. This comprehensive approach enables the achievement of complex and precise motions in the developed soft robotic systems.

**Printing performance.** To measure the printing resolution, the DLP projector was set to cover a 30.4 × 19 mm area with a 1280 × 800 pixel resolution. Under these conditions, the system demonstrated a resolution of ~100 $\mu$m (width ratio 0.94) (Supplementary Fig. 16), comparable to the previous study[40]. For the main printing process, a 3D magnetic field of >80 mT is generated within a $\phi$40 mm workspace.

Unlike Direct Ink Writing (DIW) techniques that limit magnetization to the extrusion direction (Supplementary Fig. 17a), this DLP-based approach enables spatial magnetization programming for individual pixels in thin-walled structures. The layer-by-layer curing process facilitates the fabrication of curved magnetic thin-walled structures without fully encapsulating supports, overcoming limitations in creating suspended or curved geometries (Supplementary Fig. 17b). This capability allows for complex 3D geometries with controlled magnetization in multiple directions.

Printing speed is determined by factors including resin exposure time, magnetic field switching speed, and print head movement. For a soft machine with a 200 $\mu$m layer thickness, the custom resin mix with PDMS requires a 21 s exposure time. This exposure time is consistent with other studies using similar custom formulations (~15 s)[46]. When using commercial resins (Elastic 50 A or Clear V5, FormLabs Inc), the exposure time is reduced to 5 s per 100 $\mu$m. The Halbach array operates at a rotation speed of 4.32°/s, with a maximum reorientation time of 47 s for a 180° rotation. Material compatibility tests were conducted using a custom photocurable PDMS-based polymer matrix and commercial photocurable resins (Elastic 50 A and Clear V5, FormLabs Inc.). All printing materials were infused with 10 wt% NdFeB particles, with commercial resins requiring an additional 5 wt% of fumed silica particles for stability. Multi-material printing capabilities were demonstrated through the creation of structures combining materials with different mechanical properties (Supplementary Fig. 18). This versatility enables the production of complex 3D geometries with spatially varying magnetization profiles, such as biomimetic pumps with curved surfaces (Fig. 6).

Detailed fabrication parameters, support removal information for all prototyped soft devices are provided in Supplementary Table 1.

**Post-processing.** Support structures were removed from high curvature and thin-walled three-dimensional magnetic structures using elongated precision scissors (Supplementary Fig. 19 and Supplementary Discussion 10). The soft nature of the elastomer structures allows passive deformation during support removal. This facilitated the complete removal of supports while avoiding damage to the magnetic structures. Specific support removal times are listed in Supplementary Table 1. For three-dimensional structures with enclosed cavities, additional simple assembly was performed using one-component room temperature curing silicone (705, Ausbond).

**Testing of three-dimensional magnetic field generation.** A three-dimensional Hall probe (3AHD802F, Beijing CH-HALL Electronic Devices Co., Ltd) measured the magnetic field strength and direction at the curing plane. Custom LabVIEW programs recorded the measurements. The device was calibrated by mapping the designed magnetic field values to the Halbach array rotation angle ($\alpha$) and the solenoid input current. Field strength and direction at the center of the printing plane, including the in-plane and out-of-plane components, were measured (Supplementary Fig. 1). After calibration, theoretical magnetic field values were set, and field strength and direction were measured at various points within a 40 mm diameter to evaluate accuracy and uniformity (Supplementary Fig. 2).

**Preparation of NdFeB microparticles.** Equal volumes (60 ml) of 5 $\mu$m NdFeB microparticles, zirconia grinding balls, and anhydrous ethanol were added to a 250 ml vacuum ball milling jar. The grinding balls fractured the NdFeB microparticles, with anhydrous ethanol as the grinding medium. The jar was evacuated and filled with argon to prevent oxidation. The planetary ball mill was set to 500 rpm for 3 h to expedite fracturing, then 440 rpm for another 3 h to further reduce and homogenize the particle size. Liquid nitrogen was introduced to keep the temperature below zero. After grinding, the grinding balls were removed, and the ethanol was separated in a vacuum drying oven to obtain dry 1–2 $\mu$m NdFeB microparticles (Supplementary Fig. 3).

**Magnetic characterization.** A vibrating sample magnetometer (7404, Lake Shore) measured the magnetic moment density of the printed samples for finite element simulations. The magnetic moment density of saturated magnetized samples was also measured as a control to assess the alignment quality of the magnetic particles. The samples were rectangular ($7 \times 2 \times 1$ mm$^3$) with magnetization along the thickness. The magnetic hysteresis loops were measured within an external magnetic field range of − 1.45 T to 1.45 T. The magnetic moment density was obtained by dividing the remanent magnetization by the volume. Printed samples were fabricated under an 80 mT orienting magnetic field, while saturated samples were printed under a 0 mT field and then magnetized in a ~5 T pulse field.

**Mechanical testing.** A tensile testing machine (IBTC-300SL, CARE Measurement & Control) performed quasi-static tensile tests on the samples under a 30 N load. Dogbone-shaped specimens were printed without a magnetic field, with an overall length of 24 mm, gauge length of 4 mm, and cross-sectional area of 5 × 1.6 mm$^2$. Stress-stretch curves were plotted, and the curves were fitted using a Neo-Hookean model to obtain the shear modulus ($\mu$) for finite element simulations.

**Biocompatibility testing.** To evaluate the biocompatibility of our 3D-printed magnetic soft machines, we conducted preliminary cytotoxicity tests using four sample types: (A) pure PDMS (Sylgard-184), (B) printed parts treated with a serial extraction protocol[47], (C) printed parts treated as in (B) and then coated with PDMS (Sylgard-184), and

(D) printed parts cleaned with isopropanol and coated with PDMS (Sylgard-184).

Sample preparation was carried out as follows: For type A, Sylgard-184 was mixed at a 10:1 base to curing agent ratio, degassed, and cured at 120 °C for 5 min. Printed parts (types B, C, and D) were fabricated using our UMA-SL method with a poly(dimethylsiloxane)-based resin containing magnetic particles. After printing, type B samples underwent a serial extraction protocol as described by Bhattacharjee et al.[47]. Type C samples received the same treatment as type B, followed by coating. Type D samples were cleaned with isopropanol before coating. For types C and D, a layer of PDMS (Sylgard-184) was applied to isolate the internal magnetic particles. All samples were sterilized by UV irradiation before use in cell culture experiments. The coating process and surface characterization are detailed in the Supplementary Fig. 14a.

Human umbilical vein endothelial cells (HUVEC) were used for the cytotoxicity assay. Cells were cultured in Dulbecco's modified Eagle's medium (DMEM) supplemented with 10% (v/v) fetal bovine serum and maintained at 37 °C in a humidified incubator with 5% $CO_2$. The four sample types were placed in the HUVEC culture environment, with cells cultured in medium only serving as positive controls.

To assess cell viability, we performed live/dead cell staining using Calcein AM. Cells around the test objects were sampled at 24 h intervals for a total of 72 h. After staining, the cells were washed with PBS and observed under a fluorescence microscope with an excitation wavelength of 488 nm. The resulting fluorescence microscopy images, showing cell growth around the samples over time, are presented in the Supplementary Fig. 14b.

**Finite-element analysis.** Finite element simulations (Abaqus, Dassault Systèmes) modeled deformation shapes and strain distribution of all designed models under external magnetic fields. A user-defined subroutine defined the material's constitutive model[48]. Input parameters included shape, magnetization distribution, magnetic moment density, shear modulus, and external magnetic field. The effect of magnetization direction on deformation magnitude and posture was optimized, and optimal magnetization distributions were studied. Strain distribution after deformation was analyzed to identify regions for adding support structures.

**Magnetic actuation.** Handheld permanent magnets (Neodymium52) generated magnetic fields to control shape changes of magnetic structures. Magnetic structures were placed between two opposing magnets to enhance field uniformity and reduce gradient force effects. An octupole orthogonal electromagnet device generated a rotating field to drive the capsule robot with a maximum strength of 40 mT.

## Data availability
The data generated in this study are provided in the main text, supplementary information, and source data file. Source data are provided with this paper.

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

## Acknowledgements

We thank Prof. Ying Hu and Prof. Xingwen Wang from Life Science and Technology College, Harbin Institute of Technology for their great support with the cytotoxicity test. This work was supported in part by National Key R& D Program of China under Grant 2023YFB4705600 (H.X.), and in part by National Natural Science Foundation of China under Grant 61925304 (H.X.), 62127810 (H.X.).

## Author contributions

X.M., S.L. and H.X. conceived and designed the study. X.M., S.L., and X.S. developed the UMA-SL system and fabricated the magnetic soft machines. S.L., C.T., and L.M. were responsible for characterizing the printed magnetic soft machines. X.M., S.L., and H.X. analyzed the experimental data. H.X. and X.M. provided supervision for the study. All authors contributed to writing the manuscript.

## Competing interests

The authors declare no competing interests.
