## [Transparent Peer Review file · Nature Communications]

Programmable spatial magnetization stereolithographic printing of biomimetic soft machines with thin-walled structures

Corresponding Author: Professor Hui Xie

Version 0:

Reviewer comments:

Reviewer #1

(Remarks to the Author)

This work proposes a method for DLP printing of magnetoelastic structures with supporting structures. Magnetic alignment is achieved using a Halbach array and a solenoid, and curing is accomplished with a commercial UV lamp. The thin-wall structure is facilitated by the supporting structure. While the spatial resolution of this method is discussed, it is not directly compared with existing works. The demonstrations include transporting a droplet, a heart-inspired pump and valve, and a group of soft capsules for delivering liquids. Some of these demonstrations have been shown in other works. Compared to previous studies, the main contribution of this work is proposing a method to build thin 3D structures made of magnetoelastic materials by adding supporting structures. However, the advantages of the proposed method are not clearly justified, and the demonstrated biomedical applications are highly idealized, lacking important justifications regarding magnetic field strength and device biocompatibility.

1. The combination of DLP and supporting structures seems incremental and requires justification regarding advances in printing resolution, speed, and complexity. The supporting structure in this work is key to realizing the thin-wall structure. Using supporting materials for delicate structures has been previously explored in Kim et al., *Nature* 2018. Similarly, using DLP-like methods for printing magnetoelastic materials was reported in Xu et al., *Sci. Robot.* 2019.

Kim, Y., Yuk, H., Zhao, R., Chester, S.A. and Zhao, X., 2018. Printing ferromagnetic domains for untethered fast-transforming soft materials. *Nature*, 558(7709), pp.274-279.

Xu, T., Zhang, J., Salehizadeh, M., Onaizah, O. and Diller, E., 2019. Millimeter-scale flexible robots with programmable three-dimensional magnetization and motions. *Science Robotics*, 4(29), p.eaav4494.

2. The three demonstrated applications all claim biomedical relevance, which needs careful justification as they appear impractical from a biomedical engineering perspective. Firstly, the printed material is cytotoxic, as the ferromagnetic particles in the matrix will have direct contact with cells, raising significant issues for merging the structure with the intestine. Secondly, the printed heart valve and pump share the same biocompatibility issues. Furthermore, when implanted, actuation of the pump and valve presents significant challenges.

3. The magnetic fields used in these applications are in the hundreds of milli-Tesla range, significantly higher than typical magnetic fields used in existing magnetic actuation systems. Reducing the magnetic field is not straightforward due to the necessity of sufficient magnetic particle volume for adequate torque or force. Increasing the magnetic particle concentration could be a solution, but it would also complicate the UV curing process and raise the elastic modulus of the printed structure. This issue requires careful investigation.

4. Another major concern is the reliance on removing the supporting structure. It is unclear how the supporting structure can be removed, especially for complex designs. The authors briefly mentioned using "elongated precision ophthalmic scissors" for manual removal. Experimental images of the removal process should be provided, and the applicability of the removal

method for different structures should be discussed. The authors noted in the text, "Residual supports shorter than 500 μm were observed after removal." How do these residual supports compare with the thin-wall structure, and do they deteriorate the printing precision?

5. Magnetization profiles are critical for realizing the motions. It is not clear whether these profiles are obtained by intuition or optimization. For all demonstrations, how are the magnetization profiles designed? What role does simulation play in this design process? Please elaborate.

6. For the heart-inspired pump demonstration, the coordination of the two valves is unclear. Provide time-resolved images of the pumping process.

7. In the demonstration in Fig. 5, why is the tube fixed? When the tube is not fixed, is pumping still possible? How does this relate to real-world scenarios?

8. The uniformity of layer thickness should be quantified, as experimental images (Fig. 2b) suggest that edge thickness may differ from middle parts.

9. Discuss the discrepancy between simulation and measurements in the magnetic field data shown in Fig. 1e.

Other points related to presentation:

10. Please quantify the maximum printable structure size with the current setup. Dimensions should be marked in Fig. 1.

11. Color bars should be provided for several figures, such as Fig. 3, Fig. 4, Fig. 5a, Fig. 6a, and Fig. 7a.

12. The color coding of the supporting material is somewhat confusing as it is essentially the same material as the structure.

Reviewer #2

(Remarks to the Author)

This manuscript titled "Programmable spatial magnetization stereolithographic printing of biomimetic soft machines with thin-walled structures" reports a fabrication platform for millimeter to centimeter scale soft devices that responds to external magnetic field by deformation. This fabrication strategy prints various kinds of structures by utilizing pre-magnetized magnetic particles, uniform magnetic field created at the printing workspace, and the design of supporting structures. Focus is placed upon the printing of devices with thin walls. And biomimetic devices are created such as ones that mimic human heart and human intestine. Overall, this paper is well written with beautiful schematics, and the results are nicely presented. The reviewer has the following concerns and hopefully the authors could help address them.

1. The core of this work is about a modified 3D printing fabrication strategy. However, the authors did not give common parameters that should be given, including printing resolution, printing speed, printing area, compatibility with different printing materials, possibility of multi-material printing, etc. Without explicit values for these aforementioned parameters, it is difficult to agree with the authors that this newly proposed fabrication approach is better than previously reported fabrication methods.

2. The authors presented lots of results but there lack reasons behind these experiments. Sometimes it feels like that an experiment is performed just because other people have done this experiment in their papers. And it is not clear how this experiment contributes to this specific manuscript. For example, the experiment shown in Fig. 2f shows a printed box with letters on it. In other people's work, this experiment has been used to showcase the resolution and the capability of the fabrication method to program patterns. But here, the resolution is not mentioned, so it cannot be supported by the experiment. And the letter "X", "Y", "-X", and "-Y" are obviously flawed and cannot be differentiated from each other. So, this experiment also doesn't support the claim that the proposed fabrication method has the ability to program patterns. Then, why is it here? It raises more questions than it answers. The same can be said for many other experiments reported in this manuscript.

3. The devices shown in Fig. 4c, d have been shown in a previous study (Yoonho Kim et al. Nature 2018). So, these results do not support the authors' claim that previous studies cannot fabrication thin-walled structures and this new strategy is advantageous in this aspect.

4. Two biomimetic devices are reported to mimic human intestine and human heart. These are nice ideas, but the experimental results are only preliminary. Their similarities with human organs are farfetched and not convincing. As for the soft capsules reported in Section 2.7, I don't think there is anything new here besides what have been well studied by other preceding papers.

Minor comments:

1. When "DLP" is first mentioned on Page 2, it is not defined.

2. On Page 6, "Simulations analyzed three wall thicknesses (200, 300, 400, 500 μm) and support spacings of 1-5 mm." Is

“three” a typo and it should be “four”? And what is the internal for the spacings of 1-5 mm?

3. Page 13, “Future designs of heterogeneous robots could enable the independent control of each capsule, allowing the team to release drugs at multiple target points as needed.” It is not very straightforward how “independent control” could be achieved, considering that the control of individual agent using a single external input (magnetic field or something else) is still a grand challenge in the field.

Overall, the reviewer thinks this manuscript does report an interesting approach to fabricate magnetic soft devices. But it is not clear how this new approach contributes to the state-of-the-art and whether or not it offers clear benefits in comparison with existing ones.

Version 1:

Reviewer comments:

Reviewer #1

(Remarks to the Author)

The authors have addressed most of my comments in this round of revision.

What are the noteworthy results?

- The authors proposed a method to print thin wall structures by SLA with supports, demonstrating thinner magnetic structures which have not been shown before.

Will the work be of significance to the field and related fields? How does it compare to the established literature? If the work is not original, please provide relevant references.

- Yes. This work proposes a method to build thinner magnetic structures compared with previous work particularly Kim 2018, Xu 2019. However, the authors may need to cite recent works on flexible magnetic peristaltic pumps and flexible magnetic valves which have been overlooked in the current reference list. This should be addressed.

Does the work support the conclusions and claims, or is additional evidence needed?

- The work supports the conclusions.

Are there any flaws in the data analysis, interpretation and conclusions? Do these prohibit publication or require revision?

- No.

Is the methodology sound? Does the work meet the expected standards in your field?

- Yes.

Is there enough detail provided in the methods for the work to be reproduced?

- Yes.

Reviewer #2

(Remarks to the Author)

The reviewer thanks the authors for responding to every question and comment in details. As the reviewer pointed out in the previous round of peer-review, this manuscript is well written with beautiful schematics and the results are nicely presented. The reviewer would also like to extend this praise to the Response Letter. It is very well written. However (unfortunately), the reviewer's main concern remains: the contribution of the manuscript is incremental.

In the previous round of peer-review, the reviewer raised four major questions. They are concerns about

- 1) the advantages of this method in comparison with existing fabrication methods in terms of common printing parameters;
- 2) the motivation and justification of the presented experiments which are similar with preceding studies;
- 3) the claim about previous fabrication method cannot make thin-walled structures; and
- 4) the results of the reported biomimetic devices.

The reviewer raised these four questions because the reviewer was trying to figure out what significant novelty and contribution this work has. Is it about fabrication parameters like resolution? Is it about more impressive and unprecedented experiments? Is it about totally new fabrication capabilities? Or is it about the creation of novel and useful devices? The authors' responses to these four questions are very detailed and well written. The responses to these four questions share the same cornerstone, which is that this work is trying to solve a problem that is very specific to magnetically controlled soft miniature devices. Thus, the authors argue that

- 1) although this work's fabrication parameters like resolution isn't much better, it is acceptable;
- 2) the experiments that are similar with previous studies are used to validate the feasibility of proposed fabrication method, so it is also acceptable;
- 3) although the previous work can also do thin-walled structure, the newly proposed method can achieve more variations in the magnetization direction while printing; and
- 4) the biomimetic devices are only for demonstration and validation of the fabrication method, they are not intended for clinical applications.

So, everything comes to this one question: whether this work is addressing a critical challenge in the field or not? Or two challenges in the authors' words: complex 3D structures and precise magnetization profile. If the answer is yes, then the authors' responses make sense, and the manuscript can be well justified and supported. But the reviewer's opinion is that, although there are still some limitations in 3D structures and magnetization profiles of soft miniature devices, these limitations are now more or less "tolerable" by people working in this field because most critical fabrication limitations have been removed by the large collection of wonderful articles on this topic from 2018-2021. In the field of magnetic small-scale robots, people can now fabricate most of the designs they want. In the reviewer's opinion, the remaining fabrication limitations can no longer be treated as a grand challenge. This change of the research trend can be felt by the observation that we no longer see influential articles focusing on fabrication of these devices on top-tier journals (such as Nat. Commun.) in the past three, four years. And the field of Small-scale Robotics now focuses on finding real-world applications and relevance for these miniaturized devices.

In summary, the reviewer likes this article. It is easy to read. The figures are clear and intuitive. The authors have obviously put a significant amount of effort in polishing the manuscript and the response letter. But the reviewer must admit that the reviewer still thinks this is an incremental work.

Reviewers' Comments and Authors Response

Manuscript ID: NCOMMS-24-30796

Manuscript title: Programmable spatial magnetization stereolithographic printing of biomimetic soft machines with thin-walled structures

1. Response to comments from Reviewer #1

Dear Reviewer,

Please see our statements to your valuable comments on our manuscript:

I. General Comments:

This work proposes a method for DLP printing of magnetoelastic structures with supporting structures. Magnetic alignment is achieved using a Halbach array and a solenoid, and curing is accomplished with a commercial UV lamp. The thin-wall structure is facilitated by the supporting structure. While the spatial resolution of this method is discussed, it is not directly compared with existing works. The demonstrations include transporting a droplet, a heart-inspired pump and valve, and a group of soft capsules for delivering liquids. Some of these demonstrations have been shown in other works. Compared to previous studies, the main contribution of this work is proposing a method to build thin 3D structures made of magnetoelastic materials by adding supporting structures. However, the advantages of the proposed method are not clearly justified, and the demonstrated biomedical applications are highly idealized, lacking important justifications regarding magnetic field strength and device biocompatibility.

■ **Reply:** We sincerely appreciate your comprehensive review of our manuscript and the valuable insights you have provided. We would like to address the points you raised:

1. **Contribution and advantages of our method:** The method proposed in our work introduces significant advancements in the fabrication of 3D hard-magnetic soft machines with both programmable 3D thin-walled geometries and precise 3D magnetization programming. The key innovations include:
 - (a) An advanced magnetic field generation system using Halbach arrays and a solenoid, which produces a highly uniform 3D magnetic field (>80 mT) within a $\phi 40$ mm workspace. This represents a substantial increase in the printing area compared to previous methods.
 - (b) Optimized support structure design specifically tailored to address challenges arising from magnetic field-structure interactions during printing. This enables the fabrication of complex 3D geometries with precise magnetization control, which was not achievable with previous methods.
2. **Comparison with existing works:** While we acknowledge that some demonstrations share similarities with previous studies, our method significantly advances the field by enabling the fabrication of more complex and precisely magnetized 3D structures. We have provided a detailed comparison with existing works, particularly those by Kim et al. (*Nature* 2018) and Xu et al. (*Sci. Robot.* 2019), to highlight the novelty and advantages of our approach.

3. **Spatial resolution:** We have conducted additional analyses to quantify our method's spatial resolution, achieving approximately 100 μm resolution, which is comparable to state-of-the-art techniques in the field.
4. **Biomedical applications and magnetic field strength:** We acknowledge that our demonstrated applications are at a proof-of-concept stage. To address concerns about magnetic field strength, we have optimized our structures to operate effectively at lower field strengths (30-200 mT), which are readily achievable with common magnetic sources. We have also conducted preliminary biocompatibility tests and discussed potential strategies for improving biocompatibility for future clinical applications.
5. **Device biocompatibility:** We have addressed biocompatibility concerns by conducting cytotoxicity experiments and discussing strategies for enhancing biocompatibility, such as silica encapsulation of magnetic particles and surface treatments. We recognize that further in-depth studies are required for clinical applications and have outlined our plans for future work in this direction.

We believe these clarifications and additional experiments demonstrate the significant contributions of the UMA-SL printing technique we developed to the field of 3D magnetoelastic structure fabrication. We have revised our manuscript to more clearly articulate these points and provide a more comprehensive justification for our method's advantages. Thank you again for your valuable feedback, which has helped us improve the quality and clarity of our work.

II. Major Comments:

- 1) The combination of DLP and supporting structures seems incremental and requires justification regarding advances in printing resolution, speed, and complexity. The supporting structure in this work is key to realizing the thin-wall structure. Using supporting materials for delicate structures has been previously explored in Kim et al., *Nature* 2018. Similarly, using DLP-like methods for printing magnetoelastic materials was reported in Xu et al., *Sci. Robot.* 2019.

Kim, Y., Yuk, H., Zhao, R., Chester, S.A. and Zhao, X., 2018. Printing ferromagnetic domains for untethered fast-transforming soft materials. *Nature*, 558(7709), pp.274-279.

Xu, T., Zhang, J., Salehizadeh, M., Onaizah, O. and Diller, E., 2019. Millimeter-scale flexible robots with programmable three-dimensional magnetization and motions. *Science Robotics*, 4(29), p.eaav4494.

- **Reply:** Thank you for your thoughtful comments and for giving us the opportunity to clarify the advancements our proposed UMA-SL printing method brings to the field. Before discussing the specific performance metrics of our approach, such as printing resolution, speed, and complexity, we would like to clarify the motivation behind this research to better illustrate its contributions to the field. We greatly appreciate the foundational contributions of Kim et al., *Nature* 2018^[1], and Xu et al., *Sci. Robot.* 2019^[2], which have significantly advanced the fabrication of magnetically responsive soft materials. This work seeks to address the remaining critical challenges toward a common goal shared by researchers in the field: fabricating three-dimensional (3D) hard-magnetic soft machines with both **programmable 3D geometries** and **precise 3D magnetization programming** throughout the structures.

Hard-magnetic soft materials (elastic modulus < 1 MPa), particularly those containing NdFeB particles, are integral to this goal due to their high remanence, which allows them to generate stronger actuation forces in magnetic fields. This makes them particularly suited for soft robotics applications, where large deformations are essential. In 2D structures, **thin-film** designs are often employed to minimize resistance to deformation, thus enabling **large deformations**. Similarly, when evolving from 2D to 3D structures, **thin-wall** architecture becomes essential for achieving large deformations under magnetic fields by significantly reducing bending stiffness. **This is why we focus on fabricating thin-walled soft machines, as this structure is crucial for addressing key challenges in magnetic soft robotics.** In this context, the thin-walled structure is a critical feature for 3D hard-magnetic soft materials, as illustrated in Fig. R1a.

The complexity in such work arises from two primary aspects: the fabrication of thin-walled 3D structures, such as vertical walls and curved surfaces, and the precise control of magnetization across the entire structure. Both aspects are vital for achieving the desired performance in soft machines.

To realize these complex magnetic soft machines, researchers face two major interconnected challenges:

- i) **Fabrication of complex 3D structures:** Creating thin-walled 3D geometries, such as vertical walls and curved surfaces, while maintaining the intended magnetization orientation throughout the process.
- ii) **Precise control of magnetization:** Accurately programming the magnetization orientation of magnetic particles in three dimensions during manufacturing, ensuring its stability even with the addition of subsequent layers.

Previous researchers have made significant progress in addressing these challenges. Kim et al. (2018) demonstrated the first successful 3D printing of hard-magnetic soft materials using direct

ink writing (DIW) technology. Their method achieved precise 2D magnetization control by aligning magnetic particles along the extrusion direction with a one-dimensional magnetic field, as shown in Fig. R1b i. By shielding the printing material from external fields, they avoided field-structure interactions but were limited to **2D magnetization programming**.

It's important to note that in Kim et al.'s process, the printed structures were not immediately cured but rather heated for curing after the printing was completed. **The supporting structures they used were a standard feature of DIW technology, primarily aimed at addressing the general structural instability typical in that process (Fig. R1b ii).** These supports and their overall strategy were fundamentally unrelated to the challenges we face in 3D magnetization programming for complex geometries.

Building on this, Xu et al. (2019) made an important advancement by extending magnetization programming to 3D using digital light processing (DLP) technology. They controlled the magnetization direction by rotating a permanent magnet, but their work was limited to **small-scale planar structures** (Fig. R1c i). A key challenge they identified was the difficulty of achieving a uniform magnetic field over larger areas. As Xu et al. stated, "*The magnet can be replaced by three pairs of electromagnetic coils for reorienting the magnetic particles. Electromagnetic coils can generate a uniform magnetic field in a 3D cubic workspace **without magnetic field sensors**, making it possible to expand the workspace area. **The biggest challenge** for this design is the capacity of the electromagnetic coils. Because the resin vat is nested in the coil system and the coil must provide at least **80 mT in all directions...** the system would require **powerful amplifiers, large amounts of copper wires, and an effective heat dissipation system**" (Xu et al., *Sci. Robot.* 2019).*

Crucially, true 3D magnetization programming requires the printed structure to be exposed to a magnetic field during the fabrication process, as demonstrated by Xu et al. This exposure introduces significant challenges, particularly for thin-walled 3D geometries. The interaction between the magnetic field and the structure can lead to misalignment between newly printed layers and the existing structure, potentially resulting in print failures due to improper layer adhesion or significant deviations from the intended geometry and magnetization distribution (Fig. R1c ii). These challenges highlight the difficulty of scaling their system for more thin-walled complex 3D geometries.

Our work introduces two key innovations that significantly advance the field beyond existing methods:

- i) **Advanced Magnetic Field Generation:** While previous works, such as Xu et al. (2019), achieved small-scale planar magnetization control, their systems faced substantial limitations in uniformity and scalability of the magnetic field. Our development of a Halbach array-based system overcomes these challenges by generating a highly uniform 3D magnetic field (>80 mT) within a $\phi 40$ mm workspace, increasing the printing area by more than **an order of magnitude** (Fig. R1d i), while consuming under 280 W. This allows for precise and consistent magnetization even in larger and more complex 3D geometries, addressing the critical issue of non-uniform field distribution that limited prior approaches.
- ii) **Optimized Support Structure Design:** Unlike the conventional supports used in Kim et al. (2018), which were standard for addressing general structural stability issues during DIW printing, our support structures are specifically designed to address the challenges arising from the interaction between the magnetic field and the printed structure (Fig. R1d ii, details in manuscript Fig. 3). This approach ensures precise control over magnetic particle alignment during printing, which is essential for fabricating 3D magnetic soft machines that inherently require thin-walled structures. By resolving issues like misalignment from magnetic field distortions, our method enables the successful fabrication of structures with much greater geometric and magnetic complexity than before. This represents a fundamental shift from previous methods limited to simpler 2D magnetization.

Figure R1: Comparison of 2D and 3D magnetization and structure fabrication approaches, highlighting advancements in thin-walled 3D magnetic soft machines. (a) Transition from 2D thin-film to 3D thin-walled structures for large deformations. (b) Kim et al., Nature 2018: (i) 2D magnetization with uniform magnetic fields, and (ii) conventional DIW method for printing 3D thin-walled structures. (c) Xu et al., Sci. Robot. 2019: (i) 3D magnetization with non-uniform magnetic fields, but limited to small-scale planar structures, and (ii) DLP method lacking effective support and magnetic field control for 3D thin-walled structures. (d) This work (UMA-SL): (i) 3D magnetization and structure fabrication with uniform magnetic fields and a $\phi 40$ mm printing area, and (ii) a novel support strategy to stabilize thin-walled structures against magnetic forces.

It is important to emphasize that our UMA-SL technique is not simply a combination of DLP technology and supporting structures, but rather a systematic solution to the critical challenge of precise 3D magnetization programming in complex geometries. While our prototype addresses this

Figure R2: Analysis of DLP printing resolution. (a) Microscopic images of printed lines. Top: Magnified view showing detailed width measurement. Bottom: Wider view of multiple lines with varying widths. (b) Printed vs. design line width ratio. Dashed line indicates 1:1 ratio.

fundamental issue, we consider it a significant step towards the common goal of the field, not a definitive solution. We recognize that our current system may not excel in all aspects of 3D printing performance. Our primary focus has been on advancing the crucial aspect of 3D magnetization programming in thin-walled complex geometries, while maintaining acceptable performance in other areas.

To provide a complete overview and context for this UMA-SL method, we present the current capabilities of our method in terms of printing resolution, speed, and complexity:

- i) **Printing resolution:** We set the DLP projector to a 30.4×19 mm area with a resolution of 1280×800 pixels, yielding a theoretical resolution of $23.75 \mu\text{m}$. The preset line widths ranged from 75 to $500 \mu\text{m}$. Microscopic images of the printed structures are shown in Fig. R2a. For smaller widths, the actual printed lines were narrower than the preset values due to the Gaussian light intensity distribution of the pixels^[3]. As the preset widths increased, the printed-to-preset width ratio approached 1, indicating reduced edge effects. This trend is clearly illustrated in Fig. R2b. Our achievable resolution was approximately $100 \mu\text{m}$ (width ratio 0.94), comparable to Xu et al. (*Sci. Robot.*, 2019). Future improvements could involve lens upgrades, grayscale printing, and material optimization.
- ii) **Printing speed:** The printing speed depends on the resin exposure time, magnetic field switching speed, and print head movement. For our soft machine with a $200 \mu\text{m}$ layer thickness, the custom resin mix with PDMS results in a 21-second exposure time due to unoptimized material properties, which is comparable to another study using a similar custom formulation (~ 15 seconds)^[4].
The Halbach array rotates at $4.32^\circ/\text{s}$. For the maximum 180° rotation, including particle alignment and curing initiation, the total reorientation time is under 47 seconds, slightly faster than that of Xu et al., *Sci. Robot.* 2019 (2 minutes). Typical scenarios with smaller rotations require proportionally less time. As shown in Table R1, printing time increases with magnetic field complexity (e.g., "Cube" structure), but our uniform magnetic field generator allows parallel printing of multiple units, improving efficiency (e.g., "Capsule Robots"). Future work will focus on optimizing the resin, reducing exposure time, and enhancing magnetic field switching speed to further accelerate printing.
- iii) **Printing complexity:** The evolution of magnetic soft machines from 2D to 3D introduces dual challenges: thin-walled structures and precise 3D magnetization programming within these structures. Our Halbach array-based magnetic field generator solves the issue of uniform strong magnetic field generation in the curing area, while optimized support structures (Fig. 3a-c) stabilize the structure against magnetic forces during fabrication without impeding deformation.

Table R1: Fabrication and support removal details of all prototyped soft devices.

Prototyped Devices	Batches	Units/ Batch	Layers/ Unit	Mag. Dir./ Layer	Time/ Batch (hrs)	Total Time (hrs)	Supp. Rem./ Unit (mins)	Hollow Ratios
Cube	1	1	50	2~5	5.1	5.1	0	N/A
Thin-walled tube	1	1	50	2	2.4	2.4	5	0.92
Biomimetic aortic valve	1	1	15	3	0.9	0.9	0	N/A
Biomimetic mitral valve	1	1	25	2	1.2	1.2	2	N/A
Octagonal tube	1	1	50	2	2.4	2.4	5	0.72
Multi-unit 3D lattice	1	1	15	2	0.7	0.7	5	0.53
Biomimetic colonic peristaltic machine	2	2	50	2	2.4	4.8	5	0.81
Biomimetic heart pumping machine	1	1	121	2	3.2	3.2	10	0.81
Capsule Robots	1	6	55	2	2.6	2.6	5	0.65

* **Batches:** Number of print batches. **Units/Batch:** Number of units printed per batch. **Layers/Unit:** Number of layers for each unit. **Mag. Dir./Layer:** Number of different magnetization field directions applied per layer. **Time/Batch (hrs):** Fabrication time required for each batch in hours. **Total Time (hrs):** Total fabrication time for all batches combined. **Supp. Rem./Unit (mins):** Time required to remove support structures for a single unit. **Hollow Ratio:** Ratio of internal cavity volume (V_{hollow}) to total volume of the device (V_{total}), calculated as $V_{\text{hollow}}/V_{\text{total}}$. **N/A** indicates that the hollow ratio is not applicable due to the lack of internal cavities.

Currently, only Kim et al. have demonstrated 3D thin-walled magnetic soft structures with programmed magnetization. However, as shown in Fig. R3a, their printed fiber width constitutes the wall thickness, limiting magnetization to align only with the thin-wall's tangent direction. Their DIW process, while enabling stable stacking of uncured magnetic inks, requires supports that fully encapsulate the printed structure, limiting the creation of suspended or curved geometries and potentially reducing efficiency.

In contrast, our method enables spatial magnetization programming for each pixel in thin-walled

Figure R3: Comparison of printing complexity and magnetization control in 3D thin-walled magnetic soft structures (a) 2D tangential magnetization in vertically printed thin-walled structures (Kim et al., *Nature* 2018). (b) 3D spatial magnetization in vertically printed and curved thin-walled structures enabled by DLP printing (this work:UMA-SL).

structures. The layer-by-layer curing of DLP printing allows supports that don't fully encapsulate printed structures, effectively enabling curved magnetic thin-walled structures fabrication (Fig. R3b). We've demonstrated complex 3D geometries with controlled magnetization, such as the biomimetic heart pump with curved surfaces and a spatially varying magnetization profile (Fig. 6).

We have incorporated the discussion on fundamental challenges into the third paragraph of the Introduction. The detailed printing parameters, including resolution, speed, and complexity, have been added to the Printing Performance section within the Methods of the manuscript.

2) The three demonstrated applications all claim biomedical relevance, which needs careful justification as they appear impractical from a biomedical engineering perspective. Firstly, the printed material is cytotoxic, as the ferromagnetic particles in the matrix will have direct contact with cells, raising significant issues for merging the structure with the intestine. Secondly, the printed heart valve and pump share the same biocompatibility issues. Furthermore, when implanted, actuation of the pump and valve presents significant challenges.

■ **Reply:** Thank you for your insightful comments regarding the biomedical relevance of our demonstrated applications. We appreciate your careful consideration and would like to address your concerns in detail.

First and foremost, we want to clarify that the core objective of this manuscript is to develop a method for fabricating three-dimensional (3D) hard-magnetic soft machines with both programmable 3D geometries and precise 3D magnetization programming throughout the structures. Our choice to mimic thin-walled structures common in biological organs was primarily to validate the effectiveness of our method, rather than for direct clinical application. We apologize for any expressions in the manuscript (such as "substitute" in Fig. 5a) that may have misled readers into thinking our method is ready for immediate medical use. We have corrected these phrases accordingly.

Regarding the cytotoxicity of ferromagnetic particles:

- i) **Material Selection:** We chose a poly(dimethylsiloxane)-based material as our printing substrate, rather than conventional commercial 3D printing flexible resins which are typically cytotoxic. This material has been proven biocompatible by Nirveek Bhattacharjee et al. (*Adv. Mater.*, 2018)^[5], with properties similar to Sylgard-184, which has been widely used in biomedical applications for the past two decades.
- ii) **Research on Magnetic Particle Encapsulation:** In the field of magnetic soft machines, many researchers have extensively studied biocompatibility issues. For instance, Kim et al. (*Sci. Robot.*, 2019)^[6] and Wang et al. (*Nat. Commun.*, 2022)^[7] addressed ferromagnetic toxicity by encapsulating NdFeB particles with silica. Yang et al. (*Sci. Adv.*, 2022)^[8] successfully resolved biocompatibility issues by coating silicone mixed with NdFeB particles with silicone and modifying it with hydrogel, even implanting it into a porcine model. These studies provide important references for addressing the biocompatibility of magnetic materials.
- iii) **Surface Treatment:** Following Yang et al.'s approach, we coated the printed parts with a layer of PDMS (Sylgard-184) to isolate the internal magnetic particles.
- iv) **Cytotoxicity Experiments:** We conducted preliminary cytotoxicity tests comparing four sample types:
 - (A) pure PDMS (Sylgard-184);
 - (B) printed parts treated with Nirveek Bhattacharjee et al.'s serial extraction protocol;
 - (C) printed parts treated as in (B) and then coated with PDMS (Sylgard-184);

(D) printed parts conventionally cleaned with isopropanol and then coated with PDMS (Sylgard-184).

The samples were prepared as shown in Fig. R4a i-iii. Fig. R4a iv demonstrates successful coating (thickness $\sim 56 \mu\text{m}$). To evaluate cytotoxicity, we used live/dead cell staining to detect cell viability. Human umbilical vein endothelial cells (HUVEC) were cultured in Dulbecco's modified Eagle's medium (DMEM) supplemented with 10% (v/v) fetal bovine serum. The four sample types were placed in the HUVEC culture environment, with cells cultured in medium only serving as positive controls. All cells were grown at 37°C in a humidified incubator with 5% CO_2 . Cells around the test objects were sampled every 24 hours and stained with Calcein AM. After washing with PBS, the cells were observed under a fluorescence microscope with an excitation wavelength of 488nm.

As shown in Fig. R4b, cell growth around all four sample types was similar to the control group, indicating no visible cytotoxicity. These results suggest that coating the printed parts with Sylgard-184 (as in sample D) can effectively achieve biocompatibility.

- v) Future Improvements: We recognize that these are preliminary cytotoxicity experiments. For clinical applications, more in-depth and systematic studies are required. It's important to emphasize that the challenge of biocompatibility is not unique to our research but is shared across the entire field of magnetic soft machines. We look forward to advancing this field together with other researchers to collectively address biocompatibility issues.

Regarding the biocompatibility of heart valves and pumps: We understand that this is a significant challenge in translating the technology into practical biomedical applications. Our current research is at the proof-of-concept stage, primarily aimed at demonstrating the potential of this technology.

Figure R4: Cytotoxicity test of 3D-printed material coated with Sylgard-184 (a) Surface characterization of 3D-printed material coated with Sylgard-184. (i) Schematic of the coating process of 3D-printed structures with Sylgard-184. (ii) Microscopic image of the printed material surface before coating. (iii) Microscopic image of the printed material surface after coating with Sylgard-184. (iv) Cross-sectional view after coating, showing the interface between the Sylgard-184 layer and the printed resin. (b) Cytotoxicity test results of 3D-printed material coated with Sylgard-184. Control group (Control) and different treatment groups (A: pure Sylgard-184; B: printed parts treated with extraction protocol; C: printed parts treated and then coated with Sylgard-184; D: printed parts cleaned with isopropanol and coated with Sylgard-184) observed at 24, 48, and 72 hours. Fluorescent microscopy images show cell growth around the samples.

In future work, we plan to use our developed UMA-SL method to create more biocompatible robots tailored to specific medical needs.

Concerning the challenges of post-implantation actuation: We agree that this is a complex engineering problem. Different application scenarios require different actuation solutions, which is beyond the scope of this paper. We anticipate developing appropriate actuation devices for specific medical needs in our future work.

In conclusion, our research contributes a novel method for manufacturing soft machines with complex geometries and precise magnetization. While current biocompatibility issues and actuation challenges limit immediate clinical applications, they are common across the field of magnetic soft machines. We believe our method opens new possibilities for future biomedical innovations. Our ongoing efforts will focus on improving biocompatibility, developing suitable actuation methods, and exploring potential clinical applications. Through interdisciplinary collaboration, we aim to address these shared challenges and advance the field towards practical biomedical applications.

Accordingly, relevant content on biocompatibility has been added to the Discussion section and a new 'Biocompatibility Testing' subsection in the Methods.

3) The magnetic fields used in these applications are in the hundreds of milli-Tesla range, significantly higher than typical magnetic fields used in existing magnetic actuation systems. Reducing the magnetic field is not straightforward due to the necessity of sufficient magnetic particle volume for adequate torque or force. Increasing the magnetic particle concentration could be a solution, but it would also complicate the UV curing process and raise the elastic modulus of the printed structure. This issue requires careful investigation.

■ **Reply:** We sincerely appreciate your insightful comments regarding the high magnetic field strength used in our application and the associated challenges. We have carefully considered these points and conducted additional experiments to address them.

Regarding the concern about high magnetic field strengths, we acknowledge that our initial manuscript included examples using fields up to 300 mT. To address this, we have taken several steps to optimize our structures and reduce the required field strengths: First, we explored the possibility of increasing magnetic particle concentration. However, we found that for UV-curable methods, higher concentrations of magnetic particles reduce UV light penetration depth (Fig. 2d), making multi-layer accumulation of 3D structures more challenging. This led us to focus on alternative approaches to reduce the required driving magnetic field.

We developed an optimized structure design that leverages our thin-walled 3D structure manufacturing method. This approach involves reducing wall thickness in areas of strain concentration and increasing thickness in areas of lower stress to enhance driving force. We tested three different characteristic structures:

- i) Type 1: Cylinder with outer diameter 6 mm, uniform wall thickness 500 μm , height 5 mm.
- ii) Type 2: Reduced wall thickness to 200 μm throughout.
- iii) Type 3: Wall thickness 200 μm in strain concentration areas, with additional 300 μm thick structure in lower strain areas.

Our simulations and experiments demonstrated significant differences in deformation among these structures. Under a 40 mT magnetic field, the cross-sectional areas after deformation decreased by 5%, 43%, and 56% respectively compared to the undeformed state for Types 1, 2, and 3 (Fig. R5a). To further investigate the performance of these structures, we applied magnetic fields ranging from 20 mT to 100 mT (in 20 mT increments) and observed the resulting deformations (Fig. R5b).

Figure R5: Structural design optimization and performance comparison of magnetically responsive 3D printed structures. (a) Design, simulation, and experimental results for three types of cylindrical structures under a 40 mT magnetic field. Type 1: uniform wall thickness; Type 2: reduced wall thickness; Type 3: optimized wall thickness distribution. (b) Cross-sectional deformation of the three structure types under increasing magnetic field strengths from 0 to 100 mT. (c) Contraction ratio analysis of the three structure types as a function of applied magnetic field strength.

The images of the cross-sectional deformations clearly show that different characteristic structures require significantly different driving fields to achieve similar deformations.

We also analyzed the contraction rate, defined as (volume before deformation - volume after deformation) / volume before deformation, for each structure type across the range of applied magnetic fields (Fig. R5c). This analysis revealed that our optimized structures could reduce the required driving field strength to 20% of the original while achieving the same contraction rate, effectively lowering the driving field.

Based on these results, we found that the Type 3 structure is most effective in reducing the required magnetic field strength while maintaining high deformation capability. This optimized design principle - reducing wall thickness in strain concentration areas and increasing thickness in lower stress areas - allows for enhanced driving force and reduced deformation resistance.

To directly address the concern about high magnetic fields, we have redesigned and reprinted structures to operate at lower magnetic field strengths. These new demonstrations replace the previous high-field examples in Fig. 4c and Fig. 4d, now showcasing structures operating effectively at 100 mT and 150 mT, respectively. We would like to clarify that the high-field characterization presented in Fig. 7c (up to 320 mT) was included for comprehensive material characterization and to explore the full potential of our structures. However, the actual applications demonstrated in our study use significantly lower magnetic field strengths, typically <200 mT.

Most applications in this manuscript use magnetic fields ranging from 30-200 mT, balancing performance with practicality. To address concerns about the feasibility of generating these magnetic fields, we note that while electromagnetic coil-based drive systems typically provide fields of a few to tens of mT, permanent magnets can produce fields of tens to hundreds of mT. For example, a circular permanent magnet with a diameter of 5 cm and thickness of 3 cm can produce a 30-200 mT magnetic field at distances of 1.2-4.5 cm from the magnet surface. This demonstrates that the magnetic field strengths used in our optimized designs are readily achievable with common magnetic sources, making our approach practical for real-world applications.

Moving forward, we will continue to refine our structure optimization techniques. This will involve developing more sophisticated algorithms for identifying optimal thickness distributions based on stress analysis, exploring new geometrical designs, and investigating the integration of multi-material printing to create structures with locally tailored magnetic and mechanical properties.

Accordingly, relevant content on structural design optimization has been added to Section 5: 'Structural Optimization for Reducing Actuating Magnetic Field' in the Supplementary Information.

- 4) Another major concern is the reliance on removing the supporting structure. It is unclear how the supporting structure can be removed, especially for complex designs. The authors briefly mentioned using "elongated precision ophthalmic scissors" for manual removal. Experimental images of the removal process should be provided, and the applicability of the removal method for different structures should be discussed. The authors noted in the text, "Residual supports shorter than 500 μm were observed after removal." How do these residual supports compare with the thin-wall structure, and do they deteriorate the printing precision?

- **Reply:** Thank you for your insightful comments, particularly regarding the support structure removal process. We appreciate this opportunity to provide a more detailed explanation and additional visual evidence.

In our manuscript, we briefly mentioned the use of "elongated precision scissors" for manual removal of support structures. To offer a clearer picture of our methodology, we have prepared a detailed illustration (Fig. R6a) that demonstrates our three-step support removal process: cutting off support, checking connection, and pulling out support.

Using precision scissors, we carefully cut along the edges of the support structure. The operator relies on tactile feedback to sense the position and orientation of the scissors, continuously adjusting to ensure complete disconnection of the support while avoiding damage to the thin-wall structure. Following this, a 1.5 mm diameter semi-circular slender rod is used to check for any remaining connections. This rod is inserted into the gap created by the disconnected support and slid along the edges. Its flexibility helps avoid damaging the thin-wall structure during this check. Once all connections are confirmed to be severed, the support structure is gently separated from the thin-wall structure.

Fig. R6a visually demonstrates these steps, providing a clear guide for the process we follow. This

Figure R6: Detailed illustration and experimental demonstration of the support structure removal process. (a) Schematic diagram showing the three key steps of support removal: cutting off support, checking connection, and pulling out support. The process utilizes tactile perception and a thin rod for precise control. (b) Experiment images of the actual support removal process for a complex 3D printed structure. The sequence demonstrates the use of precision scissors, the cutting process along the whole edge, connection checking with a thin rod, and final removal and trimming of support residues.

detailed explanation, coupled with the visual aid, highlights the key points that ensure successful support removal, even for practitioners new to the technique. With proper attention to these steps, the support removal times mentioned in our manuscript are consistently achievable.

Our method is applicable to both open and semi-enclosed elastomeric printed structures, including semi-enclosed curved 3D structures that represent typical complex designs. To illustrate this point, we have included Fig. R6b, which shows the support removal process for a heart-inspired pump - a prime example of a complex structure mentioned in our manuscript.

The key to this method's success lies in utilizing the passive deformation capability of the thin-wall structure. After support removal, the thin-wall structure can undergo large deformations. This deformation capability facilitates further trimming of residual supports, thereby reducing the volume of these residual supports. This process is clearly demonstrated in Fig. R6b, where you can observe the structure's deformation during the support removal process and how it aids in minimizing residual support material.

Regarding the residual supports mentioned in our manuscript (shorter than 500 μm), these are primarily distributed in non-strain-concentrated areas of the thin-wall structure (wall thickness 200 μm -600 μm). As observed in our microscopic images (Fig. 3e in the manuscript), the overall thickness of the thin-wall area connected to the support is slightly increased, while unconnected areas maintain the designed thickness. Importantly, this increased thickness in non-strain-concentrated areas does not negatively affect the deformation capability of the thin-wall structure. In fact, as

noted in the manuscript, these residual supports act as additional magnetic structures, providing extra magnetic torque (Fig. 3b in the original manuscript).

To address your question about how residual supports compare with the thin-wall structure and their impact on printing precision, we would like to emphasize three points. First, at support connection points, there is a localized increase in thickness. However, this is limited to small, discrete areas and does not significantly impact the overall structure. Second, the majority of the thin-wall structure, particularly in strain-concentrated areas crucial for functionality, maintains the designed dimensions. Finally, as mentioned in our manuscript, the residual supports enhance the overall magnetic response of the structure. This can be viewed as a beneficial feature that improves device performance without compromising essential structural integrity.

In conclusion, our method ensures printing precision where it matters most for functionality, while leveraging the advantages of residual supports for enhanced magnetic properties. We believe this approach, as described in our manuscript and further illustrated in the new figures (Fig. R6a and Fig. R6b), offers a viable and effective solution for manufacturing complex elastic magnetic structures.

Accordingly, relevant content on structural design optimization has been added to Section 6: 'Support structure removal process' in the Supplementary Information.

5) Magnetization profiles are critical for realizing the motions. It is not clear whether these profiles are obtained by intuition or optimization. For all demonstrations, how are the magnetization profiles designed? What role does simulation play in this design process? Please elaborate.

■ **Reply:** Thank you for raising this important point. We are pleased to provide a detailed explanation of our approach to designing magnetization profiles and the role of simulation in this process.

For all demonstrations presented in our manuscript, the magnetization profiles are designed using a function-based reverse design strategy, as illustrated in Fig. R7. This approach combines physical intuition with computational optimization. We begin by predefining the target deformed shape based on functional requirements. By comparing the undeformed and target shapes, we determine the required torque distribution. We then equate the required torque to magnetic torque $T_m = V_m |\mathbf{M}| |\mathbf{B}| \sin \theta$, where θ is the angle between magnetization and external field directions. An initial magnetization profile is derived based on the required magnetic torque and a predetermined external field direction.

This initial profile is then optimized according to specific task requirements. For instance, to increase deformation, we adjust θ to maximize torque during the deformation process, as demonstrated in Fig. 5b and Fig. 6c in the manuscript. To achieve controllable motion, we reduce θ to ensure a non-zero net magnetic moment along the field direction, as shown in Fig. 7c.

Simulation plays a crucial role throughout our design process. We use it first for validation, verifying whether the initial magnetization profile results in the desired deformation under the applied magnetic field. During the optimization phase, simulation helps quantify deformation magnitude, guiding us towards the optimal magnetization distribution for maximum deformation (Fig. 5b and Fig. 6c). Additionally, simulation is vital in determining appropriate locations for support structures. We generate logarithmic strain distribution maps of the deformed model, identifying strain concentration areas (defined as regions where strain exceeds 90% of the maximum strain). This information helps us avoid placing supports in these areas, thereby minimizing their impact on deformation, as illustrated in Fig. 3b.

In summary, while this approach is grounded in a systematic reverse design strategy, it does incorporate

Figure R7: Schematic illustration of the function-based inverse design strategy for magnetization profile design. The process consists of three main steps: (1) Reverse design: defining the desired deformation based on the printed structure; (2) Equivalent torque: deriving the initial magnetization profile based on the required magnetic torque; and (3) Magnetization profile optimization: refining the profile for increased deformation or controlled motion.

elements of physical intuition, particularly in the initial stages of defining target deformations and setting up the reverse design problem. The subsequent steps involve computational optimization and extensive use of simulation for validation and refinement. We believe this comprehensive approach allows for the achievement of complex and precise motions as demonstrated in the presented results.

Moving forward, we aim to further refine our inverse design strategy and optimization algorithms. We are exploring ways to incorporate machine learning techniques to potentially discover novel magnetization profiles and to handle even more complex deformation targets. These future directions will build upon the foundation established in the current work, potentially leading to even more sophisticated and adaptable soft robotic systems.

Accordingly, relevant content on magnetization profile design has been added to subsection: 'Magnetization profile design process' in Methods.

6) For the heart-inspired pump demonstration, the coordination of the two valves is unclear. Provide time-resolved images of the pumping process.

■ **Reply:** Thank you for your question regarding the coordination of the two valves in our heart-inspired pump demonstration. We are pleased to provide a detailed explanation of the valve mechanism and its coordination in the pumping process.

Our aortic valve-inspired one-way valve is designed to be controlled through two methods: switching of external magnetic fields (Fig. 4a) and fluid pressure (Fig. 6a). In the heart-inspired pump demonstration (Fig. 6d), we specifically focused on the fluid pressure control mechanism, which closely mimics the functioning of the human aortic valve. Both valves in this setup are strategically placed outside the magnetic field area, allowing their opening and closing states to be governed solely by fluid dynamics.

The operation principle of our valve is based on the response of its leaflets to fluid pressure. When fluid pressure is applied in a specific direction, the valve leaflets fold inward, creating a trifoliate opening that permits fluid passage. Conversely, when the fluid pressure is reversed, the leaflets open outward, with their edges constraining each other, effectively closing the opening and preventing fluid passage. This mechanism is clearly illustrated in Fig. R8a.

To provide a comprehensive understanding of the valve's performance, we conducted additional experiments using an injection pump to control fluid flow through the valve at speeds ranging from 0 to 0.6 mL/s. This range was chosen to effectively cover the actual flow rate range of our heart-

Figure R8: Fluid pressure-controlled unidirectional valve system inspired by heart and its working principle. (a) Open and close states of the valve under forward and reverse flow. (b) Relationship between flow velocity and valve orifice area. (c) Coordinated operation of two valves during a complete pumping cycle.

inspired pump (0.30 mL/s to 0.56 mL/s), ensuring that our observations are relevant to the pump's operating conditions. Using a micro-camera, we captured cross-sectional images of the valve under various flow conditions. The results, presented in Fig. R8b, demonstrate that the valve opens at a low flow rate of 0.2 mL/s, and its opening area gradually enlarges as the flow rate increases.

The coordinated working state of the two valves during a complete pumping cycle is depicted in Fig. R8c. The valves are installed in opposite directions to ensure unidirectional fluid flow. During the diastolic phase, as the pressure in the pump cavity decreases, liquid flows in from the outside. The left valve opens to allow fluid passage, while the right valve closes to prevent backflow from the simulated vascular network channel. In the subsequent systolic phase, the increased pressure in the pump cavity creates fluid pressure in the opposite direction. This causes the left valve to close and the right valve to open, effectively pumping fluid into the channel.

Through this coordinated operation of the two valves, we achieve efficient unidirectional fluid delivery in our heart-inspired pump demonstration.

Accordingly, relevant content on the coordination of the two valves has been added to Section 7: 'Fluid pressure-controlled valve system in heart-Inspired pump' in the Supplementary Information.

7) In the demonstration in Fig. 5, why is the tube fixed? When the tube is not fixed, is pumping still possible? How does this relate to real-world scenarios?

■ **Reply:** We sincerely appreciate your insightful questions regarding the fixed tube design in Fig. 5. Your comments have highlighted important aspects of our experimental setup, which we are pleased to clarify further:

The tube is fixed in our experimental setup primarily because the driving force for the peristaltic wave is a moving localized magnetic field. An unfixed tube would be displaced by the gradient force of this field, disrupting the peristaltic wave generation. Fixing the tube allows us to accurately measure its deformation and observe fluid movement, ensuring consistent and reliable data on the tube's performance under magnetic actuation.

Pumping with an unfixed tube would introduce significant challenges. The tube's movement would interfere with the designed peristaltic wave generation, likely resulting in reduced efficiency and inconsistent results. Our research focuses on demonstrating the reliability of our method for manufacturing magnetic thin-walled structures, rather than addressing specific biomedical applications.

Regarding real-world scenarios, our fixed-tube design has implications for implantable magnetic devices. In biological applications, magnetic implants often need to be fixed to prevent unintended effects on other organs. For example, Yang et al. (*Sci. Adv.*, 2022)^[8] proposed an implantable magnetic soft robotic bladder (MRB) fixed with a titanized polypropylene mesh, while Chen et al. (*Nat. Commun.*, 2024)^[9] developed a magnetic multilayer soft robot that attaches to gastric tissue through an adhesive film. These studies suggest that fixing a tube-like structure within a biological system could potentially be a feasible implantation method in certain cases. However, the actual applicability would depend on specific biological contexts, implant locations, and individual patient conditions. Our experimental approach provides a foundation for exploring such possibilities, but further research would be needed to determine the viability in various real-world medical scenarios.

8) The uniformity of layer thickness should be quantified, as experimental images (Fig. 2b) suggest that edge thickness may differ from middle parts.

■ **Reply:** Thank you for your insightful comment regarding the uniformity of layer thickness. We appreciate the opportunity to address this concern.

The observed non-uniformity in layer thickness is primarily due to the Gaussian light intensity distribution of individual pixels in DLP projector^[3]. This results in stronger light intensity in the center and weaker intensity at the edges of the projected area. To quantify this effect, we conducted additional analyses using a sample with 200 μm layer thickness and 10 wt% NdFeB content.

The curable depth shown in Fig. 2d is derived from the middle thickness value. We observed that the actual curing depth was 240 μm , which is naturally 1.2 times the set layer thickness. We intentionally set the curing depth to 1.2 times the printing layer thickness to ensure sufficient cross-linking quality between layers. This contributes to the structural integrity of the print despite the non-uniform thickness. With an exposure time of 21 s, we found that at the 200 μm thickness interface, the cross-linking length at the edge (0 μm position) was 83% of the center.

Our analysis of the side-wall image of the cured structure and its profile curve (Fig. R9a and R9b) clearly demonstrates this thickness variation. To improve uniformity, we implemented secondary exposure compensation in the non-uniform regions. This technique improved the cross-linking length ratio from 83% to 95%, as evidenced by our experimental results. It's important to note that while we observed this non-uniformity, it did not significantly impact the functionality or performance of our printed structures. However, we did observe that the surface quality of the printed structures was relatively poor, with clearly identifiable printing layer lines. Despite this, the printed structures presented in our paper demonstrate that this level of non-uniformity still allows for functional prints suitable for their intended use.

For applications potentially requiring even higher uniformity, multiple exposures could theoretically be employed to further refine the printing process. However, this additional step was not necessary for the structures presented in our study, as they already met the required performance criteria, despite the visible layer lines.

Accordingly, relevant content on uniformity of layer thickness has been added to Section 8: 'Layer thickness uniformity analysis' in the Supplementary Information.

Figure R9: Improvement of layer thickness uniformity through secondary exposure. (a) Side-wall images of cured structures showing the original single exposure result (top) and the improved result after secondary exposure (bottom). The width of the cured region increases from 83% to 95% of the projected area width after secondary exposure. (b) Curable depth profiles across the width of the projected area. The original single exposure (orange line) shows non-uniformity, with the edge regions having a lower curable depth. After secondary exposure (red line), the uniformity is greatly improved, with the curable depth at the edges increasing to match that of the central region more closely. The dashed line indicates the target layer thickness

9) Discuss the discrepancy between simulation and measurements in the magnetic field data shown in Fig. 1e.

■ **Reply:** Thank you for your careful review of our manuscript and for bringing attention to the discrepancy between simulation and measurements in the magnetic field data shown in Fig. 1e.

We appreciate your insightful comment and would like to address this issue in detail.

In Fig. 1e, we characterized the magnetic field strength and angle error within a 40mm diameter printable range using COMSOL simulation and measurements from a three-dimensional Hall sensor to demonstrate the field uniformity in this region. The simulation data, derived from the X and Y axes within the printable range, showed a field strength range of 82.3 mT to 85.4 mT and an angle error range of 0° to 0.7° . The measurement data, collected at 10 mm intervals along the X and Y axes, revealed a field strength range of 80.0 mT to 83.4 mT and an angle error range of 0.1° to 0.9° .

We observed that while the magnetic field strength showed a notable difference between simulation and measurement, the shape of the data curves remained similar, with an offset of approximately 2 mT. The simulated magnetic field angle errors were generally smaller than the measured values, though both maintained relatively low values.

The consistency in the shape of the magnetic field strength curves between simulation and measurement suggests that the discrepancy primarily stems from a mismatch between the magnetic properties of the actual permanent magnets used and the material properties set in the simulation. Real-world factors such as material quality, manufacturing processes, installation, and environmental conditions may have contributed to a slight degradation in the magnetic performance of the actual magnets.

To verify this hypothesis, we recalibrated the material properties in our simulation, adjusting the remanent flux density from 1.44 T to 1.40 T to better reflect the characteristics of the permanent magnets used. After this adjustment, the updated simulation results showed a magnetic field strength range of 80.0 mT to 83.4 mT (Fig. R10a) and an angle error range of 0° to 0.4° (Fig. R10b). These revised results demonstrate improved consistency with the measured data, particularly in terms of magnetic field strength.

While the material property mismatch appears to be the primary factor contributing to the observed discrepancy, we acknowledge that other elements such as sensor measurement errors and assembly tolerances in the magnetic field generation device may also play a role.

We believe that this explanation adequately addresses the discrepancy noted in Fig. 1e. The adjusted simulation results maintain similar trends and ranges to the measured data, supporting the validity of our main conclusions. In future studies, we plan to further refine our simulation

Figure R10: Comparison of measured, simulated, and revised simulation data for magnetic field characteristics. (a) Magnetic field strength $|B|$ and (b) Angle error as a function of distance along the X and Y axes within the printable size. Black dots and blue dots represent measured points for field strength and angle error, respectively. Solid lines show the original simulation results, while dashed lines indicate the revised simulation after adjusting the remanent flux density. The circular insets illustrate the printable area and measurement axes.

parameters to more accurately model real-world conditions and consider using higher precision measurement equipment to minimize such discrepancies.

Accordingly, relevant content on the discrepancy between simulation and measurements of magnetic field data has been added to Section 9: 'Magnetic Field Discrepancy Analysis' in the Supplementary Information.

III. Other points related to presentation:

10) Please quantify the maximum printable structure size with the current setup. Dimensions should be marked in Fig. 1.

■ **Reply:** We are grateful for your notification. The maximum printable structure size with the current setup has a diameter of 40 mm, which has been marked in Fig. 1e.

11) Color bars should be provided for several figures, such as Fig. 3, Fig. 4, Fig. 5a, Fig. 6a, and Fig. 7a.

■ **Reply:** We appreciate your suggestion. Color bars have been added to Fig. 3, Fig. 4, Fig. 5a, Fig. 6a, and Fig. 7a in the revised manuscript to enhance data interpretation.

12) The color coding of the supporting material is somewhat confusing as it is essentially the same material as the structure.

■ **Reply:** Thank you for your comment. The supporting material, although made from the same resin, is colored differently from the thin-walled structure for clarity and to emphasize that it will be removed later. To prevent misunderstanding, we have added '(same resin)' after 'Support' in the relevant figures.

Reference

1. Kim, Y., Yuk, H., Zhao, R., Chester, S. A., & Zhao, X. (2018). Printing ferromagnetic domains for untethered fast-transforming soft materials. *Nature*, **558**(7709), 274-279.
2. Xu, T., Zhang, J., Salehizadeh, M., Onaizah, O., & Diller, E. (2019). Millimeter-scale flexible robots with programmable three-dimensional magnetization and motions. *Science Robotics*, **4**(29), eaav4494.
3. Wang, Y., Wang, Y., Mao, C., & Mei, D. (2023). Printing depth modeling, printing process quantification and quick-decision of printing parameters in micro-vat polymerization. *Materials & Design*, **227**, 111698.
4. Wu, P., Yu, T., & Chen, M. (2023). Magnetically-assisted digital light processing 4D printing of flexible anisotropic soft-Magnetic composites. *Virtual and Physical Prototyping*, **18**(1), e2244924.
5. Bhattacharjee, N., Parra-Cabrera, C., Kim, Y. T., Kuo, A. P., & Folch, A. (2018). Desktop-stereolithography 3D-printing of a poly (dimethylsiloxane)-based material with sylgard-184 properties. *Advanced materials*, **30**(22), 1800001.
6. Kim, Y., Parada, G. A., Liu, S., & Zhao, X. (2019). Ferromagnetic soft continuum robots. *Science robotics*, **4**(33), eaax7329.

7. Wang, T., Ugurlu, H., Yan, Y., Li, M., Li, M., Wild, A. M., ... & Sitti, M. (2022). Adaptive wireless millirobotic locomotion into distal vasculature. *Nature communications*, **13**(1), 4465. **13**(1), 4465 (2022).
8. Yang, Y., Wang, J., Wang, L., Wu, Q., Ling, L., Yang, Y., ... & Zang, J. (2022). Magnetic soft robotic bladder for assisted urination. *Science advances*, **8**(34), eabq1456.
9. Chen, Z., Wang, Y., Chen, H., Law, J., Pu, H., Xie, S., ... & Yu, J. (2024). A magnetic multi-layer soft robot for on-demand targeted adhesion. *Nature Communications*, **15**(1), 644.

2. Response to comments from Reviewer #2

Dear Reviewer,

Please see our statements to your valuable comments on our manuscript:

I. General Comments:

This manuscript titled “Programmable spatial magnetization stereolithographic printing of biomimetic soft machines with thin-walled structures” reports a fabrication platform for millimeter to centimeter scale soft devices that responds to external magnetic field by deformation. This fabrication strategy prints various kinds of structures by utilizing pre-magnetized magnetic particles, uniform magnetic field created at the printing workspace, and the design of supporting structures. Focus is placed upon the printing of devices with thin walls. And biomimetic devices are created such as ones that mimic human heart and human intestine. Overall, this paper is well written with beautiful schematics, and the results are nicely presented. The reviewer has the following concerns and hopefully the authors could help address them.

Overall, the reviewer thinks this manuscript does report an interesting approach to fabricate magnetic soft devices. But it is not clear how this new approach contributes to the state-of-the-art and whether or not it offers clear benefits in comparison with existing ones.

- **Reply:** We sincerely thank you for your positive evaluation and constructive comments on our research. We are greatly encouraged by your assessment that our paper “reports an interesting approach to fabricate magnetic soft devices” and your praise that it is “well written with beautiful schematics, and the results are nicely presented”.

We fully understand and appreciate your concerns regarding how our method contributes to the state-of-the-art and whether it offers clear benefits compared to existing approaches. Our work addresses critical challenges in fabricating 3D hard-magnetic soft machines with programmable geometries and precise 3D magnetization. We have made key technological breakthroughs and achieved significant improvements in system performance metrics.

In our subsequent detailed replies, we will provide comprehensive explanations, experimental data, and analytical results for each specific comment to fully support our claims of innovation and demonstrate the advantages of our method over existing technologies.

We believe that these additional explanations will address your questions and demonstrate the significance and novelty of our research. Thank you again for your insightful comments, which have helped us to improve our manuscript.

II. Major Comments:

1) The core of this work is about a modified 3D printing fabrication strategy. However, the authors did not give common parameters that should be given, including printing resolution, printing speed, printing area, compatibility with different printing materials, possibility of multi-material printing, etc. Without explicit values for these aforementioned parameters, it is difficult to agree with the authors that this newly proposed fabrication approach is better than previously reported fabrication methods.

■ **Reply:** We sincerely appreciate your insightful comments and the opportunity to clarify the advancements this study brings to the field. While we acknowledge the importance of specific performance metrics such as printing resolution, speed, area and material compatibility, we believe it is crucial to first elucidate the fundamental motivation and unique contributions of the presented research. This context will provide a clearer framework for understanding the significance of the technical parameters and innovations described.

The method proposed in our work builds upon the groundbreaking contributions of Kim et al., *Nature* 2018^[1], and Xu et al., *Sci. Robot.* 2019^[2], which have significantly advanced the fabrication of magnetically responsive soft materials. These two approaches remain the primary methods for printing magnetically responsive ferromagnetic soft materials to date. However, despite these advancements, critical challenges remain in achieving a long-standing goal shared by researchers in the field: the fabrication of three-dimensional (3D) hard-magnetic soft machines that simultaneously possess **programmable 3D geometries** and **precise 3D magnetization programming** throughout their structures. Our 3D printing methodology directly addresses these challenges, offering unique capabilities that differentiate it from previous methods in fabricating magnetically responsive soft materials.

To fully address the reviewer's concerns and demonstrate the advancements of this study, we will first elaborate on the critical challenges in the field and how our innovations directly address them. Then, we will provide detailed information on our system's current capabilities in terms of printing resolution, speed, area, and material compatibility.

a) Fundamental Challenges and Our Innovative Solutions

Hard-magnetic soft materials (elastic modulus < 1 MPa), particularly those containing NdFeB particles, are integral to this goal due to their high remanence, which allows them to generate stronger actuation forces in magnetic fields. This makes them particularly suited for soft robotics applications, where large deformations are essential. In 2D structures, **thin-film** designs are often employed to minimize resistance to deformation, thus enabling **large deformations**. Similarly, when evolving from 2D to 3D structures, **thin-wall** architecture becomes essential for achieving large deformations under magnetic fields by significantly reducing bending stiffness. **This is why we focus on fabricating thin-walled soft machines, as this structure is crucial for addressing key challenges in magnetic soft robotics.** In this context, the thin-walled structure is a critical feature for 3D hard-magnetic soft materials, as illustrated in Fig. R11a.

- i) **Fabrication of complex 3D structures:** Creating thin-walled 3D geometries, such as vertical walls and curved surfaces, while maintaining the intended magnetization orientation throughout the process.
- ii) **Precise control of magnetization:** Accurately programming the magnetization orientation of magnetic particles in three dimensions during manufacturing, ensuring its stability even with the addition of subsequent layers.

Figure R11: Comparison of 2D and 3D magnetization and structure fabrication approaches, highlighting advancements in thin-walled 3D magnetic soft machines. (a) Transition from 2D thin-film to 3D thin-walled structures for large deformations. (b) Kim et al., Nature 2018: (i) 2D magnetization with uniform magnetic fields, and (ii) conventional DIW method for printing 3D thin-walled structures. (c) Xu et al., Sci. Robot. 2019: (i) 3D magnetization with non-uniform magnetic fields, but limited to small-scale planar structures, and (ii) DLP method lacking effective support and magnetic field control for 3D thin-walled structures. (d) This work: (i) 3D magnetization and structure fabrication with uniform magnetic fields and a $\phi 40$ mm printing area, and (ii) a novel support strategy to stabilize thin-walled structures against magnetic forces.

The complexity in such work arises from two primary aspects: the fabrication of thin-walled 3D structures, such as vertical walls and curved surfaces, and the precise control of magnetization across the entire structure. Both aspects are vital for achieving the desired performance in soft

machines.

To realize these complex magnetic soft machines, researchers face two major interconnected challenges:

Previous researchers have made significant progress in addressing these challenges. Kim et al. (2018) demonstrated the first successful 3D printing of hard-magnetic soft materials using direct ink writing (DIW) technology. Their method achieved precise 2D magnetization control by aligning magnetic particles along the extrusion direction with a one-dimensional magnetic field, as shown in Fig. R11b i. By shielding the printing material from external fields, they avoided field-structure interactions but were limited to **2D magnetization programming**.

It's important to note that in Kim et al.'s process, the printed structures were not immediately cured but rather heated for curing after the printing was completed. **The supporting structures they used were a standard feature of DIW technology, primarily aimed at addressing the general structural instability typical in that process (Fig. R11b ii).** These supports and their overall strategy were fundamentally unrelated to the challenges we face in 3D magnetization programming for complex geometries.

Building on this, Xu et al. (2019) made an important advancement by extending magnetization programming to 3D using digital light processing (DLP) technology. They controlled the magnetization direction by rotating a permanent magnet, but their work was limited to **small-scale planar structures** (Fig. R11c i). A key challenge they identified was the difficulty of achieving a uniform magnetic field over larger areas. As Xu et al. stated, "*The magnet can be replaced by three pairs of electromagnetic coils for reorienting the magnetic particles. Electromagnetic coils can generate a uniform magnetic field in a 3D cubic workspace **without magnetic field sensors**, making it possible to expand the workspace area. **The biggest challenge** for this design is the capacity of the electromagnetic coils. Because the resin vat is nested in the coil system and the coil must provide at least **80 mT in all directions...** the system would require **powerful amplifiers, large amounts of copper wires, and an effective heat dissipation system**" (Xu et al., *Sci. Robot.* 2019).*

Crucially, true 3D magnetization programming requires the printed structure to be exposed to a magnetic field during the fabrication process, as demonstrated by Xu et al. This exposure introduces significant challenges, particularly for thin-walled 3D geometries. The interaction between the magnetic field and the structure can lead to misalignment between newly printed layers and the existing structure, potentially resulting in print failures due to improper layer adhesion or significant deviations from the intended geometry and magnetization distribution (Fig. R11c ii). These challenges highlight the difficulty of scaling their system for more thin-walled complex 3D geometries.

Our work introduces two key innovations that significantly advance the field beyond existing methods:

- i) **Advanced Magnetic Field Generation:** While previous works, such as Xu et al. (2019), achieved small-scale planar magnetization control, their systems faced substantial limitations in uniformity and scalability of the magnetic field. Our development of a Halbach array-based system overcomes these challenges by generating a highly uniform 3D magnetic field (>80 mT) within a $\phi 40$ mm workspace, increasing the printing area by more than **an order of magnitude** (Fig. R11d i), while consuming under 280 W. This allows for precise and consistent magnetization even in larger and more complex 3D geometries, addressing the critical issue of non-uniform field distribution that limited prior approaches.
- ii) **Optimized Support Structure Design:** Unlike the conventional supports used in Kim et al. (2018), which were standard for addressing general structural stability issues during DIW printing, our support structures are specifically designed to address the challenges arising from the interaction between the magnetic field and the printed structure (Fig. R11d ii, details in manuscript Fig.

3). This approach ensures precise control over magnetic particle alignment during printing, which is essential for fabricating 3D magnetic soft machines that inherently require thin-walled structures. By resolving issues like misalignment from magnetic field distortions, our method enables the successful fabrication of structures with much greater geometric and magnetic complexity than before. This represents a fundamental shift from previous methods limited to simpler 2D magnetization.

b) Specific Performance Metrics

Having explained the fundamental challenges and our innovative solutions, we will now provide specific details on the printing parameters mentioned by the reviewer. It's important to note that while our current prototype may not excel in all aspects of 3D printing performance, it represents a significant advancement in addressing the crucial challenge of 3D magnetization programming in thin-walled complex geometries. The following sections will detail our system's current capabilities in terms of printing resolution, speed, area, and material compatibility.

- i) Printing resolution: We set the DLP projector to a 30.4×19 mm area with a resolution of 1280×800 pixels, yielding a theoretical resolution of $23.75 \mu\text{m}$. The preset line widths ranged from 75 to $500 \mu\text{m}$. Microscopic images of the printed structures are shown in Fig. R12a. For smaller widths, the actual printed lines were narrower than the preset values due to the Gaussian light intensity distribution of the pixels^[3]. As the preset widths increased, the printed-to-preset width ratio approached 1, indicating reduced edge effects. This trend is clearly illustrated in Fig. R12b. Our achievable resolution was approximately $100 \mu\text{m}$ (width ratio 0.94), comparable to Xu et al. (*Sci. Robot.*, 2019). Future improvements could involve lens upgrades, grayscale printing, and material optimization.
- ii) Printing area and speed: Our system achieves a highly uniform 3D magnetic field (>80 mT) within a $\phi 40$ mm workspace, increasing the printing area by more than **an order of magnitude** compared to previous works such as Xu et al. (2019). This expanded workspace allows for precise and consistent magnetization even in larger and more complex 3D geometries, addressing the critical issue of non-uniform field distribution that limited prior approaches.

Printing speed depends on the resin exposure time, magnetic field switching speed, and print head movement. For our soft machine with a $200 \mu\text{m}$ layer thickness, the custom resin mix with PDMS results in a 21-second exposure time due to unoptimized material properties, which is comparable

Figure R12: Analysis of DLP printing resolution. (a) Microscopic images of printed lines. Top: Magnified view showing detailed width measurement. Bottom: Wider view of multiple lines with varying widths. (b) Printed vs. design line width ratio. Dashed line indicates 1:1 ratio.

Table R2: Fabrication and support removal details of all prototyped soft devices.

Prototyped Devices	Batches	Units/ Batch	Layers/ Unit	Mag. Dir./ Layer	Time/ Batch (hrs)	Total Time (hrs)	Supp. Rem./ Unit (mins)	Hollow Ratios
Cube	1	1	50	2~5	5.1	5.1	0	N/A
Thin-walled tube	1	1	50	2	2.4	2.4	5	0.92
Biomimetic aortic valve	1	1	15	3	0.9	0.9	0	N/A
Biomimetic mitral valve	1	1	25	2	1.2	1.2	2	N/A
Octagonal tube	1	1	50	2	2.4	2.4	5	0.72
Multi-unit 3D lattice	1	1	15	2	0.7	0.7	5	0.53
Biomimetic colonic peristaltic machine	2	2	50	2	2.4	4.8	5	0.81
Biomimetic heart pumping machine	1	1	121	2	3.2	3.2	10	0.81
Capsule Robots	1	6	55	2	2.6	2.6	5	0.65

* **Batches:** Number of print batches. **Units/Batch:** Number of units printed per batch. **Layers/Unit:** Number of layers for each unit. **Mag. Dir./Layer:** Number of different magnetization field directions applied per layer. **Time/Batch (hrs):** Fabrication time required for each batch in hours. **Total Time (hrs):** Total fabrication time for all batches combined. **Supp. Rem./Unit (mins):** Time required to remove support structures for a single unit. **Hollow Ratio:** Ratio of internal cavity volume (V_{hollow}) to total volume of the device (V_{total}), calculated as $V_{\text{hollow}}/V_{\text{total}}$. **N/A** indicates that the hollow ratio is not applicable due to the lack of internal cavities.

to another study using a similar custom formulation (~ 15 seconds)^[4]. Using commercial resin (Elastic 50 A or Clear V5) (Fig. R13) could reduce the exposure time to 5 seconds per 100 μm .

The Halbach array rotates at $4.32^\circ/\text{s}$. For the maximum 180° rotation, including particle alignment and curing initiation, the total reorientation time is under 47 seconds, slightly faster than that of Xu et al., *Sci. Robot.* 2019 (2 minutes). Typical scenarios with smaller rotations require proportionally less time. As shown in Table R2, printing time increases with magnetic field complexity (e.g., “Cube” structure), but our uniform magnetic field generator allows parallel printing of multiple units, improving efficiency (e.g., “Capsule Robots”). Future work will focus on optimizing the resin, reducing exposure time, and enhancing magnetic field switching speed to further accelerate printing.

iii) Printing material compatibility: Regarding material compatibility and the possibility of multi-material printing, we have conducted additional experiments to validate our method’s versatility. We fabricated hinged rectangular flaps with three different elastomeric components, each with oppositely directed magnetization towards both ends (Fig. R13a). The materials used include:

- Our proposed photocurable PDMS-based polymer matrix
- Commercial flexible photocurable resin (Elastic 50 A, FormLabs Inc.)
- Commercial rigid photocurable resin (Clear V5, FormLabs Inc.)

All printing materials were infused with 10 wt% NdFeB particles. The two commercial resins required an additional 5 wt% of fumed silica particles to enhance their stability.

We created three types of rectangular flaps (Fig. R13b):

- Type 1: Entirely made of photocurable PDMS structure
- Type 2: Made of flexible commercial resin

- Type 3: Upper half made of rigid resin, lower half made of photocurable PDMS

Type 3 demonstrates our method's capability for multi-material printing. We first printed the lower photocurable PDMS structure, cleaned it with isopropanol to remove uncured resin, then changed the resin container to print the upper rigid resin structure.

When subjected to a 20 mT external magnetic field, these flaps exhibited varying deformations (Fig. R14c). Type 2 showed significantly less deformation than Type 1, due to the higher Young's modulus of the flexible resin (~ 300 MPa^[5]) compared to our photocurable PDMS (421 kPa). Type 3 demonstrated a deformation similar in magnitude to Type 1 but with a different shape: Type 1 formed a narrow 'C' shape, while Type 3 formed a 'V' shape. This difference is attributed to the rigid resin (> 2 GPa^[5]) arms in Type 3 primarily transferring torque rather than deforming, concentrating the deformation at the hinge.

For the commercial resins, we maintained a consistent exposure time of 5 seconds per 100 μm layer thickness. Our photocurable PDMS-based polymer matrix was printed using the printing parameters described in the main manuscript.

These results clearly demonstrate our method's compatibility with various materials and its capability for multi-material printing, addressing the reviewer's concerns about material versatility and printing possibilities.

We have incorporated the discussion on fundamental challenges into the third paragraph of the Introduction. The detailed printing parameters, including resolution, speed, area, material compatibility, and multi-material printing capabilities, have been added to the 'Printing performance' section within the Methods of the manuscript.

Figure R13: Material compatibility and multi-material printing demonstration. (a) Schematic of the hinged rectangular flap with dimensions (in mm) and magnetization direction (\mathbf{M} , red arrows) relative to the applied magnetic field (\mathbf{B} , blue arrows). (b) Cross-sectional view of three types of printed structures using different materials: Type 1 (This work, gray) made entirely of our photocurable PDMS, Type 2 (Elastic 50A, blue) made of commercial flexible resin, and Type 3 (combination of This work and Clear V5, yellow) demonstrating multi-material printing. (c) Deformation of the three types under a 20 mT external magnetic field, showing distinct bending behaviors: Type 1 forms a narrow 'C' shape, Type 2 exhibits minimal bending, and Type 3 forms a wider 'V' shape due to the rigid upper arms.

2) The authors presented lots of results but there lack reasons behind these experiments. Sometimes it feels like that an experiment is performed just because other people have done this experiment in their papers. And it is not clear how this experiment contributes to this specific manuscript. For example, the experiment shown in Fig. 2f shows a printed box with letters on it. In other people's work, this experiment has been used to showcase the resolution and the capability of the fabrication method to program patterns. But here, the resolution is not mentioned, so it cannot

be supported by the experiment. And the letter “X”, “Y”, “-X”, and “-Y” are obviously flawed and cannot be differentiated from each other. So, this experiment also doesn’t support the claim that the proposed fabrication method has the ability to program patterns. Then, why is it here? It raises more questions than it answers. The same can be said for many other experiments reported in this manuscript.

■ **Reply:** We sincerely appreciate your thorough review and insightful comments. We understand your concerns regarding the rationale behind our experiments and their contributions to the manuscript. We would like to address these points and provide additional context for our research.

Our study addresses the dual challenges in the evolution of magnetic soft machines from 2D to 3D structures: fabricating thin-walled structures and achieving precise 3D magnetization programming within these structures (Fig. R11a). These challenges are crucial for realizing large deformations and complex motions in soft robotic applications. Our approach offers several key innovations, including a Halbach array-based magnetic field generator, optimized support structures, spatial magnetization programming for each pixel in thin-walled structures, and utilization of DLP printing’s layer-by-layer curing to fabricate curved magnetic thin-walled structures (Fig. R11d). While our experiments may appear similar to characterization tests found in other studies, they are specifically designed to validate our novel method for fabricating high-freedom and complex thin-walled 3D structures. This approach has not been previously reported in existing research on magnetic programmable structures manufactured by photocuring printing.

Regarding Fig. 2f, it demonstrates our spatial magnetization programming capability. The cube model in this experiment has specific magnetization patterns on each face, with letters indicating their magnetization direction (outward normal to each plane). The apparent indistinctness of the letters “X”, “Y”, “-X”, and “-Y” is not a manufacturing defect, but a result of the model design. To avoid structural interference between magnetized letters, their thickness was set to 1mm. The remaining parts outside the letters were divided into upper and lower sections, with the upper half magnetized in the direction opposite to “Z” and the lower half opposite to “-Z” (Fig. R14a). This design makes the “Z” and “-Z” patterns clear on the top and bottom surfaces but interferes with the recognition of “X”, “Y”, “-X”, and “-Y”. This interference is not a manufacturing flaw but a demonstration of our precise control over complex magnetization patterns.

Figure R14: Demonstration of spatial magnetization programming in a 3D printed cube structure. (a) Schematic diagram of the 3D printed cube with programmed magnetization patterns on each face. (b) COMSOL simulation results showing the magnetic field distribution on each face of the cube. The letters X, Y, and Z indicate the magnetization direction normal to each face. Dashed white lines outline the programmed letter patterns.

To further validate our results, we conducted COMSOL simulations, particularly for the cube model in Fig. 2f. The simulation results show patterns similar to those observed with the magneto-optical camera, confirming the reliability and precision of our fabrication method (Fig. R14b). In conclusion, our research not only addresses the fabrication of thin-walled 3D magnetic soft structures but also provides a method for precisely controlling the magnetization distribution within these structures. Each experiment was designed to validate the effectiveness and superiority of this method. We believe these explanations clarify the purpose, innovation, and importance of our research.

Additionally, we have added the COMSOL simulation results (Fig. R14) to the Supplementary Information to further illustrate the spatial magnetization programming of the cube structure.

3) The devices shown in Fig. 4c, d have been shown in a previous study (Yoonho Kim et al. *Nature* 2018). So, these results do not support the authors' claim that previous studies cannot fabrication thin-walled structures and this new strategy is advantageous in this aspect.

■ **Reply:** We appreciate your insightful comments and understand the concern.

As we elaborated in our response to comment 1, while Kim et al. (2018) made significant progress using DIW technology, their method was limited to 2D magnetization programming within the printing plane, with magnetization direction constrained to the extrusion direction (Fig. R11b i). As shown in Fig. R15a, Kim et al.'s printed fiber width constituted the wall thickness, limiting magnetization to align only with the thin-wall's tangent direction. Their DIW process required supports that fully encapsulated the printed structure, limiting the creation of suspended or curved geometries.

In contrast, our method enables spatial magnetization programming for each pixel in thin-walled structures. The layer-by-layer curing of DLP printing allows supports that don't fully encapsulate printed structures, effectively enabling curved magnetic thin-walled structures fabrication (Fig. R15b). We've demonstrated complex 3D geometries with controlled magnetization, such as the biomimetic heart pump with curved surfaces and a spatially varying magnetization profile (Fig. 6). It's important to note that in Kim et al.'s process, the supporting structures were a standard feature of DIW technology, primarily addressing general structural instability (Fig. R11b ii) and were unrelated to the challenges of 3D magnetization programming for complex geometries.

Figure R15: Comparison of printing complexity and magnetization control in 3D thin-walled magnetic soft structures (a) 2D tangential magnetization in vertically printed thin-walled structures (Kim et al., *Nature* 2018). (b) 3D spatial magnetization in vertically printed and curved thin-walled structures enabled by DLP printing (this work).

To better illustrate our method's advantages, we have redesigned the structures in Fig. 4c and 4d:

Figure 4c now shows two adjacent octagonal tube structures with magnetization direction perpendicular to the curing plane - a configuration unachievable with Kim et al.'s method (Fig. R16a). The opposing magnetization directions in upper and lower parts of each tube and between adjacent tubes result in alternating concave and convex deformations, creating an undulating surface.

Figure 4d presents a multi-unit honeycomb structure, where each hexagonal unit is divided into 4 magnetization zones with magnetization directions at angles to the printed structure edges (Fig. R16b). Our method allows free adjustment of these angles, enabling more complex deformations and effectively reducing the required driving magnetic field. Under an external magnetic field, the structure undergoes uniaxial stretching deformation, with each unit exhibiting consistent behavior, demonstrating the uniformity of our printing process over large areas.

Accordingly, Fig. R15 has been added to the Supplementary Information, and the comparison discussion with Kim et al. has been included in the Methods section. Figure R15 have been incorporated into the main text.

Figure R16: Redesigned 3D printed magnetic soft structures with programmable deformations. (a) Two adjacent octagonal tube structures with magnetization perpendicular to the curing plane, demonstrating alternating concave and convex deformations. (b) Multi-unit honeycomb structure with complex magnetization patterns, exhibiting uniaxial stretching deformation under an external magnetic field.

4) Two biomimetic devices are reported to mimic human intestine and human heart. These are nice ideas, but the experimental results are only preliminary. Their similarities with human organs are farfetched and not convincing. As the for soft capsules reported in Section 2.7, I don't think there is anything new here besides what have been well studied by other preceding papers.

■ **Reply:** Thank you for your insightful comments on our manuscript. We have carefully considered your feedback and have addressed each point as follows:

We would like to clarify that the core objective of this manuscript is to develop a method for fabricating three-dimensional (3D) hard-magnetic soft machines with both programmable 3D geometries

and precise 3D magnetization programming throughout the structures. Our choice to mimic thin-walled structures common in biological organs was primarily to validate the effectiveness of our method, rather than for direct clinical application.

In this study, we mimicked simplified shapes and basic motions of organs, specifically achieving simple peristaltic movement of a biomimetic intestine and basic pumping action of a heart-like structure. These designs were mainly used to showcase the feasibility of our manufacturing method, not to deeply replicate the complex functions of organs or for immediate medical use. We have revised our manuscript to more accurately reflect the current stage of our research and clarify our claims regarding similarities to human organs.

Regarding the soft capsules reported in Section 2.7, we appreciate your perspective and would like to highlight several aspects of our approach that we believe contribute new knowledge to the field:

Magnetic soft capsules have been widely studied for their wireless navigation and targeted drug delivery capabilities. Our method achieves precise fabrication of capsules with specific magnetization angles through accurate external magnetic fields. This results in controllable radial net magnetic moments, which not only drive active capsule movement but also adjust capsule orientation during drug release, ensuring the opening is vertical for efficient content discharge (as shown in Fig. 7d).

Moreover, due to magnetic field inhomogeneity $<5\%$ and angle error $<1^\circ$ within a 40 mm diameter printing range, we can simultaneously print six capsules, significantly improving manufacturing efficiency.

Through accurate magnetization angle programming, we manufactured capsules with different net magnetic moments to respond to varying strengths of driving magnetic fields, enabling independent control and multi-region targeted drug delivery. The net magnetic moment of the capsule is $|\mathbf{M}_{\text{net}}| = |\mathbf{M}| \times \cos \alpha$, where α is half the angle between magnetization directions. We set α to 75° , 65° , and 55° respectively, resulting in three capsules with different radial net magnetic moments ($|\mathbf{M}_{\text{net}1}| < |\mathbf{M}_{\text{net}2}| < |\mathbf{M}_{\text{net}3}|$), as shown in Fig. R17a.

To demonstrate independent control and multi-region drug delivery (Fig. R17b), we conducted the following experiment:

- i) We applied a 5 mT, 1 Hz rotating magnetic field to control capsule 3 (with the largest net magnetic moment) to move to its target position. We then activated the capsule using a permanent magnet to release its internal fluid, simulating drug delivery. After completing the delivery, we controlled it to leave the working environment. During this process, the other two capsules remained stationary as the magnetic torque they experienced at this field strength was less than the static friction.
- ii) We then applied an 8 mT, 1 Hz rotating magnetic field to control capsule 2, repeating the same process while capsule 1 remained stationary.
- iii) Finally, we applied a 15 mT, 0.2 Hz magnetic field to control capsule 1, repeating the process.

This experiment demonstrated our ability to independently control multiple capsules and achieve multi-region drug release, showcasing the precision and versatility of our method.

We have revised our manuscript to include these experimental details and to more clearly articulate the innovative aspects of the method proposed in our work. Specifically, these revisions have been incorporated into the Supplementary Information section 10, titled 'Independent control of multiple capsule robots'. We believe these revisions more accurately present the scope and contributions of our research.

Figure R17: Magnetization-programmed soft capsules for multi-region targeted delivery. (a) Schematic of three soft capsules with different magnetization angles and resulting net magnetic moments. (b) Time-lapse sequence demonstrating independent control and sequential payload release of three magnetization-programmed soft capsules under varying magnetic field strengths.

III. Minor comments:

1) When “DLP” is first mentioned on Page 2, it is not defined.

■ **Reply:** Thank you for pointing this out. We have now defined "DLP" on its first mention on page 2 as "digital light processing (DLP)".

2) On Page 6, “Simulations analyzed three wall thicknesses (200, 300, 400, 500 μm) and support spacings of 1-5 mm.” Is “three” a typo and it should be “four”? And what is the internal for the spacings of 1-5 mm?

■ **Reply:** Thank you for your careful review. 'Three' is indeed a typo; we analyzed four wall thicknesses (200, 300, 400, 500 μm). The support spacings ranged from 1 mm to 5 mm (with 1 mm intervals). We appreciate your attention to detail, and these points have been corrected in the revised manuscript.

3) Page 13, “Future designs of heterogeneous robots could enable the independent control of each capsule, allowing the team to release drugs at multiple target points as needed.” It is not very straightforward how “independent control” could be achieved, considering that the control of individual agent using a single external input (magnetic field or something else) is still a grand challenge in the field.

■ **Reply:** We appreciate the reviewer’s important question regarding the challenge of independently controlling multiple robotic units using a single external input. We agree that this is indeed a significant challenge in the field.

As detailed in our response to major comment 4, we have conducted supplementary experiments to explore this challenge and demonstrate initial progress in this direction. These experiments demonstrate our ability to achieve a degree of independent control and multi-region targeted drug delivery through precise magnetization angle programming. By manufacturing capsules with different net magnetic moments, we were able to make them respond to varying strengths of driving magnetic fields. This approach allowed us to independently control multiple capsules and achieve multi-region drug release in an ideal environment, showcasing the precision and versatility of our method. The details of these experiments and results have been added to the Supplementary Information section 10 'Independent control of multiple capsule robots'.

In this new section, we also acknowledge the limitations of this approach, particularly in complex real-world applications. Consequently, we have revised our manuscript to present a more cautious and accurate view of future prospects:

“Future designs of heterogeneous robots will continue to explore methods for achieving more refined control of each capsule. While fully independent control of multiple units through a single external input remains a significant challenge, our preliminary research lays the foundation for future precise drug release at multiple target points. However, we recognize that achieving this goal in complex real-world applications will require further innovations and breakthroughs.”

Reference

1. Kim, Y., Yuk, H., Zhao, R., Chester, S. A., & Zhao, X. (2018). Printing ferromagnetic domains for untethered fast-transforming soft materials. *Nature*, **558**(7709), 274-279.
2. Xu, T., Zhang, J., Salehizadeh, M., Onaizah, O., & Diller, E. (2019). Millimeter-scale flexible robots with programmable three-dimensional magnetization and motions. *Science Robotics*, **4**(29), eaav4494.
3. Wang, Y., Wang, Y., Mao, C., & Mei, D. (2023). Printing depth modeling, printing process quantification and quick-decision of printing parameters in micro-vat polymerization. *Materials & Design*, *227*, 111698.
4. Wu, P., Yu, T., & Chen, M. (2023). Magnetically-assisted digital light processing 4D printing of flexible anisotropic soft-Magnetic composites. *Virtual and Physical Prototyping*, *18*(1), e2244924.
5. Li, Z., & Diller, E. (2022, July). Multi-material fabrication for magnetically driven miniature soft robots using stereolithography. In *2022 International Conference on Manipulation, Automation and Robotics at Small Scales (MARSS)* (pp. 1-6). IEEE.

Reviewers' Comments and Authors Response

Manuscript ID: NCOMMS-24-30796A

Manuscript title: Programmable spatial magnetization stereolithographic printing of biomimetic soft machines with thin-walled structures

1. Response to comments from Reviewer #1

Dear Reviewer,

Please see our statements to your valuable comments on our manuscript:

I. General Comments:

The authors have addressed most of my comments in this round of revision.

What are the noteworthy results?

- The authors proposed a method to print thin wall structures by SLA with supports, demonstrating thinner magnetic structures which have not been shown before.

Will the work be of significance to the field and related fields? How does it compare to the established literature? If the work is not original, please provide relevant references.

- Yes. This work proposes a method to build thinner magnetic structures compared with previous work particularly Kim 2018, Xu 2019. However, the authors may need to cite recent works on flexible magnetic peristaltic pumps and flexible magnetic valves which have been overlooked in the current reference list. This should be addressed.

Does the work support the conclusions and claims, or is additional evidence needed?

- The work supports the conclusions.

Are there any flaws in the data analysis, interpretation and conclusions? Do these prohibit publication or require revision?

- No.

Is the methodology sound? Does the work meet the expected standards in your field?

- Yes.

Is there enough detail provided in the methods for the work to be reproduced?

- Yes.

■ Reply:

We sincerely thank you for your thorough review and positive assessment of our manuscript. We are grateful for your recognition of our work's novelty, particularly in achieving thinner magnetic structures through SLA printing with supports.

Following your recommendation, we have incorporated relevant references including recent works on flexible magnetic pumping and valving applications ([30] Han et al., *IEEE Trans. Ind. Electron.*, 2023; [31] Sun et al., *Nat. Commun.*, 2024; [32] Dong et al., *Sci. Adv.*, 2024) in the Introduction section. Specifically, in the first paragraph of the Introduction where we discuss the advantages of magnetic actuation, we have revised the text to : "...**These materials facilitate programmable,**

three-dimensional deformations in small-scale soft robots[29], enhancing their functionality in tasks requiring precise manipulation, effective locomotion[12,28], pumping and valving[30-32], or greater actuation forces[2,27]...". This addition provides a more comprehensive context for our work while highlighting how our thin-walled structure fabrication method could potentially benefit these applications.

We are pleased that you find our methodology sound and reproducible, and that our data and analysis adequately support our conclusions.

Thank you again for your valuable feedback which has helped improve the quality of our manuscript.

2. Response to comments from Reviewer #2

Dear Reviewer,

Please see our statements to your valuable comments on our manuscript:

I. General Comments:

The reviewer thanks the authors for responding to every question and comment in details. As the reviewer pointed out in the previous round of peer-review, this manuscript is well written with beautiful schematics and the results are nicely presented. The reviewer would also like to extend this praise to the Response Letter. It is very well written. However (unfortunately), the reviewer's main concern remains: the contribution of the manuscript is incremental.

In the previous round of peer-review, the reviewer raised four major questions. They are concerns about

- 1) the advantages of this method in comparison with existing fabrication methods in terms of common printing parameters;
- 2) the motivation and justification of the presented experiments which are similar with preceding studies;
- 3) the claim about previous fabrication method cannot make thin-walled structures; and
- 4) the results of the reported biomimetic devices.

The reviewer raised these four questions because the reviewer was trying to figure out what significant novelty and contribution this work has. Is it about fabrication parameters like resolution? Is it about more impressive and unprecedented experiments? Is it about totally new fabrication capabilities? Or is it about the creation of novel and useful devices? The authors' responses to these four questions are very detailed and well written. The responses to these four questions share the same cornerstone, which is that this work is trying to solve a problem that is very specific to magnetically controlled soft miniature devices. Thus, the authors argue that

- 1) although this work's fabrication parameters like resolution isn't much better, it is acceptable;
- 2) the experiments that are similar with previous studies are used to validate the feasibility of proposed fabrication method, so it is also acceptable;
- 3) although the previous work can also do thin-walled structure, the newly proposed method can achieve more variations in the magnetization direction while printing; and
- 4) the biomimetic devices are only for demonstration and validation of the fabrication method, they are not intended for clinical applications.

So, everything comes to this one question: whether this work is addressing a critical challenge in the field or not? Or two challenges in the authors' words: complex 3D structures and precise magnetization profile. If the answer is yes, then the authors' responses make sense, and the manuscript can be well justified and supported. But the reviewer's opinion is that, although there are still some limitations in 3D structures and magnetization profiles of soft miniature devices, these limitations are now more or less "tolerable" by people working in this field because most critical fabrication limitations have been removed by the large collection of wonderful articles on this topic from 2018-2021. In the field of magnetic small-scale robots, people can now fabricate most of the designs they want. In the reviewer's opinion, the remaining fabrication limitations can no longer be treated as a grand challenge. This change of the research trend can be felt by the observation that we no longer see influential articles focusing on fabrication of these devices on top-tier journals

(such as *Nat. Commun.*) in the past three, four years. And the field of Small-scale Robotics now focuses on finding real-world applications and relevance for these miniaturized devices.

In summary, the reviewer likes this article. It is easy to read. The figures are clear and intuitive. The authors have obviously put a significant amount of effort in polishing the manuscript and the response letter. But the reviewer must admit that the reviewer still thinks this is an incremental work.

■ **Reply:** We greatly appreciate your thorough review and constructive comments. We are especially grateful for your recognition that the paper is “well written with beautiful schematics” and “easy to read with clear and intuitive figures”. Indeed, we invested considerable effort in making our figures and presentation clear and intuitive for two important reasons:

1. To effectively communicate the fundamental principles and advantages of our method to the broader scientific community. As the field evolves from 2D to 3D magnetic soft machines, thin-walled structures with programmable magnetization become essential for achieving large deformations. We believe clear visualization of these concepts is crucial for advancing the field.
2. To make our fabrication approach accessible to researchers beyond the manufacturing community, particularly those in biomedical and soft robotics fields. When these researchers envision novel applications requiring complex 3D magnetic soft structures, they should be able to easily understand whether our method could help realize their designs.

We understand your concern about the incremental nature of our contribution, particularly in the context of the field’s evolution. While significant progress has been made during 2018-2021, we respectfully suggest that our work provides an integrated solution to crucial remaining challenges in fabricating 3D hard-magnetic soft machines. Our key technological breakthroughs include:

1. Integration Challenge: Our method uniquely combines a scalable uniform magnetic field generation system ($\phi 40$ mm workspace, over 30 times larger than previous work) with novel support structures specifically designed for magnetic field-structure interactions. This integration enables efficient fabrication of complex thin-walled structures with precise 3D magnetization control.
2. Process Reliability: Our approach offers improved process reliability and reproducibility, as demonstrated through:
 - Compatibility with multiple materials (including PDMS-based and commercial photocurable resins)
 - Capability for multi-material printing
 - Potential for selective actuation through precise magnetization control
3. Application Potential: While our biomimetic demonstrations are at a proof-of-concept stage, they validate our method’s capability in creating complex thin-walled magnetic structures with controlled deformation. These preliminary results, though not fully developed for immediate application, confirm our method’s versatility for advancing magnetic soft robotics.

We acknowledge that the field has shifted focus toward practical applications, as you astutely observed. We view our work as contributing to this transition by providing a more reliable and integrated fabrication approach that could help accelerate the development of practical applications. Currently, we are collaborating with researchers in biological and medical fields, and we look forward to expanding these interdisciplinary collaborations to further explore the potential of our fabrication method in addressing real-world challenges.

We sincerely thank you for helping us improve our manuscript and better position our work within the field’s current context.